# A seismic analysis of subglacial lake D2 (Subglacial Lake Cheongsuk) beneath David Glacier, Antarctica

Hyeontae Ju[1,2], Seung-Goo Kang[3], Yeonjin Choi[3], Sukjoon Pyun[2], Min Je Lee[3], Hoje Kwak[4], Kwansoo Kim[1], Yeadong Kim[5], Jong Ik Lee[3]

[1]Center of Technology Development, Korea Polar Research Institute, Incheon 21990, Korea
[2]Department of Energy Resource Engineering, Inha University, Incheon 22212, Korea
[3]Division of Glacier & Earth Sciences, Korea Polar Research Institute, Incheon 21990, Korea
[4]Unit of Antarctic Inland Research, Korea Polar Research Institute, Incheon 21990, Korea
[5]Korea National Committee on Polar Research, Incheon 21990, Korea

*Correspondence to:* Seung-Goo Kang (ksg9322@kopri.re.kr)

**Abstract.** Subglacial lakes beneath Antarctic glaciers are pivotal in advancing our understanding of cryosphere dynamics, basal hydrology, and microbial ecosystems. We investigate the internal structure and physical properties of Subglacial Lake D2 (SLD2), which is located beneath David Glacier in East Antarctica, using seismic data acquired during the 2021/22 austral summer. The dataset underwent a comprehensive processing workflow, including noise attenuation, velocity analysis, and prestack time migration. The migrated seismic sections revealed distinct reverse-polarity reflections at the glacier–lake interface; however, reflections from the lake–bed sediment interface were ambiguous, leading to interpretational uncertainty about the presence of a sediment layer. To resolve this interpretational uncertainty, two alternative structural models were established: Model 1 (no sediment) and Model 2 (with a sediment layer). Synthetic seismograms generated by wave-propagation modeling were compared with field data to validate the subglacial lake structure. The results confirmed the water column thickness to be approximately 82 m (Model 1) or approximately 10 m (Model 2), and possible structural scenarios for the subglacial lake were presented. Additionally, discontinuous reflections detected in seismic sections transverse to the ice flow were interpreted as scour-like feature surfaces formed by ice movement. This study identified the basal structure beneath the subglacial lake, which had been challenging to identify with conventional radar surveys, through seismic surveying. In addition, ambiguous signals in the field seismic data were mitigated via quantitative comparison with synthetic data, thereby facilitating interpretation of the underlying structure. Collectively, these findings enhance our understanding of subglacial lake environments and inform the selection of future drilling sites for in situ sampling.

## 1 Introduction

Subglacial lakes beneath the Antarctic ice sheet are typically overlain by glaciers several kilometers thick and have remained isolated from direct atmospheric and solar influences for millions of years, creating extreme environments characterized by low temperatures (Thoma et al., 2010) and high pressures (Tulaczyk et al., 2014). With increasing scientific interest, subglacial

lakes have become a focal point for studies related to the Antarctic paleoclimate, as inferred from lake sediments, as well as
investigations into microbial life in polar ecosystems (Bell et al., 2007, 2011; Bentley et al., 2009; Christner et al., 2014;
Engelhardt et al., 1990; Priscu and Christner, 2003; Rose, 1979; Wingham et al., 2006). Subglacial lakes in Antarctica are
generally categorized as either stable or active. Approximately 80% of subglacial lakes in Antarctica are classified as stable
subglacial lakes. These closed systems do not exhibit significant surface elevation changes and are characterized by long-term
balance between recharge and discharge, although the extent of subglacial water exchange remains uncertain in the absence of
direct observations. The remaining 20% are classified as active subglacial lakes, which exhibit surface elevation changes due
to episodic water drainage and refilling events (Livingstone et al., 2022). Such active lakes can reduce basal friction as they
expand, thereby facilitating glacier flow and, in some cases, accelerating calving processes, ultimately influencing glacier
dynamics (Bell et al., 2007; Stearns et al., 2008; Winsborrow et al., 2010). Characterizing subglacial lakes is essential for
understanding cryospheric processes, reconstructing past climate conditions, and assessing the potential for life in isolated,
extreme environments.
The sampling of subglacial lake water, sediments, and microbial communities is critical to address these scientific objectives.
However, successful sampling requires careful selection and characterization of the drilling site. Airborne ice-penetrating radar
(IPR) surveys are commonly employed at regional scales to detect potential subglacial lakes suitable for drilling (Christianson
et al., 2012; Lindzey et al., 2020; Yan et al., 2022). However, due to signal attenuation in water, IPR surveys are limited in
resolving the internal structure of subglacial lakes. To overcome this limitation, seismic surveys have been conducted at
potential subglacial lake candidates identified from IPR surveys. During such surveys, P-waves propagate through the water
column and are partially reflected at the lake–bed interface because of contrasts in acoustic impedance. Analyzing these
reflected waves enables detailed delineation of the water column and underlying substrate, thereby informing optimal drilling
locations (Brisbourne et al., 2023; Filina et al., 2008; Horgan et al., 2012; Woodward et al., 2010).
As such, numerous studies have utilized seismic surveys to investigate the characteristics of subglacial lakes, including
Subglacial Lake Ellsworth, Subglacial Lake Whillans, and Subglacial Lake CECs. Subglacial Lake Ellsworth, located beneath
2,930–3,280 m of glacial ice in West Antarctica, was the subject of a seismic survey during the austral summer of 2007–08.
This survey revealed spatially variable ice thickness and a lake water column ranging from 52 to 156 m, which guided the
identification of an optimal drilling location (Smith et al., 2018; Woodward et al., 2010). Subglacial Lake Whillans lies beneath
approximately 800 m of ice. Seismic observations conducted during the 2010/11 field season revealed water columns
extending over a 5 km segment of the survey profile, with a maximum thickness of less than 8 m. The glacier bed was
predominantly composed of soft sediments, and localized zones with shallow water columns (< 2 m) were also identified
(Horgan et al., 2012). Subsequent drilling in the summer of 2012/13 confirmed the presence of microbial life in both the water
and sediment samples (Christner et al., 2014). Subglacial Lake CECs (SLCECs), located beneath 2653 m of ice at the Rutford–
Institute–Minnesota Divide in West Antarctica, were investigated through seismic surveys conducted in the 2016/17 and
2021/22 seasons. These surveys revealed a maximum water column thickness of 301.3 ± 1.5 m and clastic sediments up to 15

m thick covering the lakebed. While the lake center was relatively flat, significant topographic variability was observed near the lake margins (Brisbourne et al., 2023).

We have initiated subglacial lake research beneath David Glacier, the closest major glacier to Jang Bogo Station in East Antarctica. Satellite altimetry has identified six subglacial lakes in this region (Smith et al., 2009; Wright and Siegert, 2012). During the 2016/17 austral summer, an airborne IPR survey was conducted over the region encompassing Subglacial Lake D1 (SLD1) and Subglacial Lake D2 (SLD2) (Lindzey et al., 2020). A subsequent high-resolution IPR survey was carried out during the 2018/19 field season, focusing solely on SLD2 (also referred to as "Subglacial Lake Cheongsuk") (Ju et al., 2025). Ju et al. (2025) subdivided the previously identified single subglacial water body at SLD2, as detected by ICESat altimetry, into three smaller subglacial lakes: SLD2-A, SLD2-B, and SLD2-C. Among these, SLD2-A represents the largest areal extent, and targeted seismic surveys were conducted over this area to obtain high-resolution information on the lake depth and basal structure. In the 2019/20 season, an initial seismic campaign identified the glacier thickness and suggested the presence of the lake; however, the data quality was compromised by surface crevasse noise and a lack of adequate fold coverage, limiting detailed interpretation. A refined seismic survey with 8-fold coverage was conducted during the 2021/22 season to address these issues. Furthermore, the sound source was positioned further from the crevasse (end-shot), delaying the arrival of crevasse-generated noise and preventing it from obscuring key reflections.

In this study, we present a detailed analysis of the physical and structural properties of SLD2-A using seismic data acquired during the 2021/22 campaign. We first describe the seismic data processing workflow, including noise attenuation, amplitude correction, and prestack time migration. Some areas of the processed field seismic data are challenging to interpret due to a lack of subsurface information, overlap with ghost signals, and signal attenuation. In the case of the SLD2 region, the absence of borehole data introduces inherent uncertainty into the subglacial lake structure derived from the Prestack Time Migration (PSTM) section. In particular, reflections associated with the sediment layer are challenging to interpret because they have weak amplitudes and overlap with ghost components. To compensate for these limitations, a subsurface structural model was constructed, and model-based synthetic seismograms were compared and analyzed against field observations. As a result, the substructure of SLD2-A is quantitatively presented as two possible scenarios: Glacier–Lake–Bedrock (model 1) or Glacier–Lake–Sediment–Bedrock (model 2).

## 2 Subglacial Lake D2 Beneath David Glacier in Antarctica

### 2.1 David Glacier

David Glacier, located in Victoria Land, East Antarctica, originates from the Dome C and Talos Dome regions and flows seaward through the Drygalski Ice Tongue (Fig. 1). The mass balance of glaciers from 1979 to 2008 has been estimated at 7.5 $\pm$ 0.4 Gt yr$^{-1}$ (Rignot et al., 2019), while the mean ice discharge over the more extended period from 1979 to 2017 was reported to be approximately 9.7 Gt yr$^{-1}$ (Frezzotti et al., 2000; Rignot et al., 2019). According to Smith et al. (2020), satellite altimetry observations from ICESat-1 and ICESat-2 (2003–2019) indicate that the grounded portion of David Glacier experienced a mass

gain of 3 ± 2 Gt yr⁻¹, whereas the adjacent ice shelves exhibited a mass loss of –1.6 ± 1 Gt yr⁻¹. Although the overall mass
balance of David Glacier currently appears stable, several active subglacial lakes observed by satellites have the potential to
influence glacier dynamics (Ju et al., 2025; Kim et al., 2025).

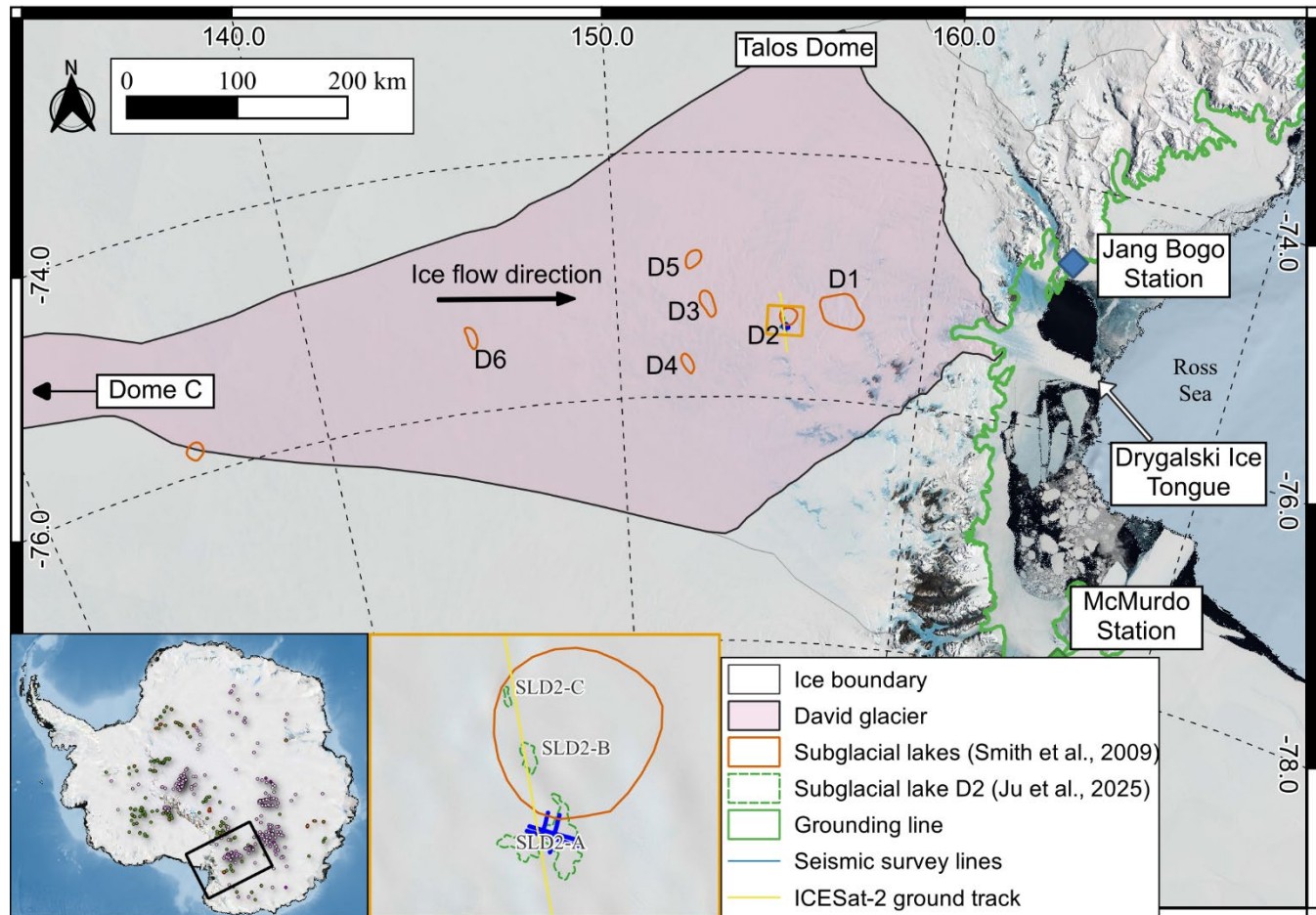

**Figure 1: Locations of subglacial lakes D1–D6 in the David Glacier region, Victoria Land, Antarctica (EPSG: 4326–WGS84).**

## 2.2 Subglacial Lake D2

Among the six subglacial lakes (D1–D6) identified beneath David Glacier via satellite altimetry (Smith et al., 2009; Wright
and Siegert, 2012), SLD2 was observed to have experienced a drainage event between 2003 and 2008 on the basis of ICESat
altimetry data (Smith et al., 2009). Since the drainage event, a continuous increase in surface elevation over SLD2 has been
observed, indicating water refilling, as detected from CryoSat-2 altimetry data (2013–2017) (Siegfried and Fricker, 2018) and,
more recently, from ICESat-2 observations (2019–2024) (Fig. 2). Figure 2 shows elevation changes relative to April 2019,
indicating surface uplift through January 2022. After this period, the surface elevation remained stable in the region originally
delineated as SLD2 by Smith et al. (2009), whereas a decreasing elevation trend was observed in the SLD2-A region (Ju et al.,
2025). These patterns of elevation change strongly suggest that SLD2 is an active subglacial lake, and that such drainage and
refilling are likely contributing to the presence of subglacial sediments (Siegfried et al., 2023).

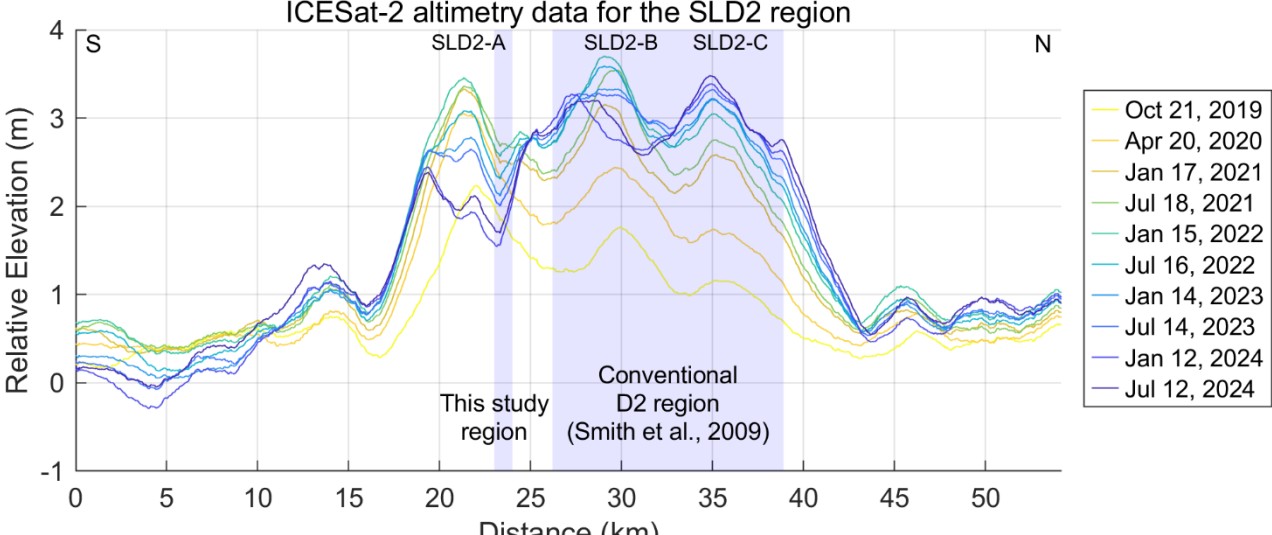


**Figure 2: Glacier surface elevation changes derived from ICESat-2 altimetry between 22 April 2019 and 12 July 2024. The X-axis corresponds to the 22 April 2019 dataset, and all subsequent elevation changes are referenced to this date. The light blue shaded region indicates the spatial overlap between the conventional SLD2 region identified by Smith et al. (2009) and our study region.**


To better constrain the extent and basal conditions of SLD2, we used airborne IPR data collected during the 2016/17 (Lindzey
et al., 2020) and 2018/19 (Ju et al., 2025) field campaigns. These surveys show that the glacier surface elevations in the SLD2
region range from approximately 1820 to 1940 m. The corresponding ice thicknesses vary between 1685 and 2293 m.
Furthermore, the observations of moderately enhanced radar bed echoes relative to the surrounding area, elevated specularity
values (>0.4), depressed basal elevations (≤–350 m), the presence of a basin-like topography, a lower hydraulic head than the
surroundings, and low hydraulic gradients (≤ 0.84°) collectively suggest a high potential for the presence of subglacial water
beneath SLD2. (Ju et al., 2025; Lindzey et al., 2020).

## 3 Method

### 3.1 Seismic survey

As previously noted, the internal structure and water column of subglacial lakes cannot be fully resolved using IPR alone because of signal attenuation in water. Accordingly, a seismic survey was conducted within the candidate SLD2-A region identified from IPR data to investigate the structure of the subglacial lake more precisely.

During the 2019/20 austral summer, a preliminary seismic survey was conducted over the SLD2-A region to evaluate the potential presence of a subglacial lake and to obtain initial information on its structural characteristics. Owing to limited field time and equipment constraints, the fold of coverage for all survey lines was restricted to 1, and all shot points happened to be aligned near surface crevasses. Consequently, the acquired seismic data were significantly degraded by strong linear coherent noise generated by crevasses, severely compromising the quality of key reflectors, particularly those at the subglacial lake–bedrock interface. Furthermore, explosives were deployed in shallow boreholes (< 20 m depth), and due to the absence of proper backfilling, poor coupling between the explosives and the borehole walls further reduced energy transmission efficiency, resulting in overall low-quality reflection signals (Ju et al., 2024). Combined with the limitations of single-fold acquisition, stacking was not feasible, the dataset exhibited a low signal-to-noise ratio (SNR) and was unsuitable for quantitative structural interpretation. Nevertheless, the preliminary survey qualitatively confirmed the glacier thickness beneath SLD2-A and suggested the presence of subglacial water, providing critical baseline information that guided the methodology and survey design of the subsequent detailed seismic campaign conducted during the 2021/22 season.

For the refined survey, seismic acquisition lines were planned using bed topography derived from the IPR and surface elevation data from satellite altimetry. A total of four seismic lines were acquired and designated 21X, 21Y, 21XX, and 21YY (Fig. 3). Lines 21X and 21XX, oriented approximately 60° relative to the ice flow direction, are situated at an average surface elevation of 1894 ± 13 m. Lines 21Y and 21YY, oriented approximately -30° in the ice flow direction, lie at an average elevation of 1887 ± 16 m. All lines traverse regions of minimal topographic relief, with average surface slopes of approximately 0.5°, indicating a relatively flat and stable glacier surface. The lengths of the 21X/21XX and 21Y/21YY lines are approximately 5 km and 3.5 km, respectively. Seismic acquisition for lines 21X and 21Y was conducted using 8-fold coverage to increase the resolution, whereas lines 21XX and 21YY were acquired with 4-fold coverage due to time constraints during the survey. The additional acquisition parameters are summarized in Table 1.

Table 1: Parameters of the active-source seismic survey.

| Survey Parameters | Survey lines | | | |
|---|---|---|---|---|
| | 21X line | 21Y line | 21XX line | 21YY line |
| Line length (km) | 5 | 3.5 | 5 | 3.5 |
| Fold | 8 | 8 | 4 | 4 |
| Shot interval (m) | 90 | 90 | 180 | 180 |
| Number of shots | 56 | 40 | 28 | 20 |

| Shot positioning | Use both off-end and center shots |
|---|---|
| Receiver channels | 96 |
| Receiver interval (m) | 15 |
| Near offset (m) | 0 |
| Far offset (m) | 1425 |
| Recording time (s) | 4 |
| Record peak frequency (kHz) | 1 |
| Record sampling rate (ms) | 0.25 |
| Survey time (days) | 34 |
| Survey crew size | Hot water drilling (3), Seismic (6) |

155

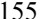

**Figure 3: 21/22 seismic survey layout (black lines) overlaid on (a) bed elevation and (b) hydraulic head data from IPR results (Ju et al., 2025).**

Before the seismic survey, a ground-penetrating radar (GPR) survey was used to identify the firn transition zone at depths of approximately 20–22 m. To enhance seismic signal transmission, 1.6 kg of pentaerythritol tetranitrate (PETN) explosives were emplaced at depths of 25–30 m using hot water drilling techniques. A total of 144 shots were deployed across the four survey lines. Detailed shot positioning information is provided in the supplementary information S1. Given the snow-covered glacier surface, Georods were used instead of conventional spike-type geophones to increase signal detection efficiency (Voigt et al., 2013). Each Georod houses four geophone elements in a 0.6 m-long cylindrical array, producing a single output by summing the inputs from all the elements. Compared with traditional geophones, this configuration improves coupling and detection performance in snow-dominated environments (Voigt et al., 2013). Figure 4 presents shot gather #27 from line 21X and shot gather #7 from line 21Y. In these shot gathers, both the direct wave and the refracted wave velocities were derived from first-arrival travel-time analysis. The direct wave velocity was estimated to be approximately 1800 m s$^{-1}$, while the higher-velocity arrival—interpreted as a refracted wave traveling through the firn–ice transition zone—exhibited an apparent velocity of approximately 3800 m s$^{-1}$. First-arrival analysis of the direct wave indicates a normal polarity, confirming the source waveform polarity. A prominent negative polarity reflection is observed at a two-way travel time (TWT) of approximately 1.2 s, interpreted as the glacier–lake interface (①; See Table 2 for symbols definitions). Approximately 25–30 ms later, a ghost reflection (②) with normal polarity appears. A subsequent reflection at approximately 1.3 s TWT, showing normal polarity, is attributed to the bed interface (⑤), followed by its negative polarity ghost reflection (⑥) 25–30 ms later. In some shot gathers, reflection signal (③) and its corresponding ghost signal (⑤) are observed. Notably, while signal ③ generally appears with normal polarity in most records, it appears with reverse polarity in a few cases, such as Shot #27 on line 21X. The survey was designed to place the seismic source at a distance from crevasses, ensuring that crevasse-related noise would be recorded after the main reflections (1.1–1.3 s), thereby minimizing its impact (Figure 4a). While most data exhibit crevasse noise occurring after the main reflections, a reduction in the source–crevasse distance causes this noise to increasingly overlap with the primary arrivals, thereby complicating interpretation.

**Table 2: Symbols for each reflection event**

| Interface symbols | Model 1 | Model 2 |
|:---:|---|---|
| ① | Ice-water | Ice-water |
| ② | Ice-water ghost | Ice-water ghost |
| ③ | - | Water-sediment |
| ④ | - | Water-sediment ghost |
| ⑤ | Water-bed | Sediment-bed |
| ⑥ | Water-bed ghost | Sediment-bed ghost |
| ⑦ | Ice-bed | Ice-sediment |
| ⑧ | Ice-bed ghost | Ice-sediment ghost |

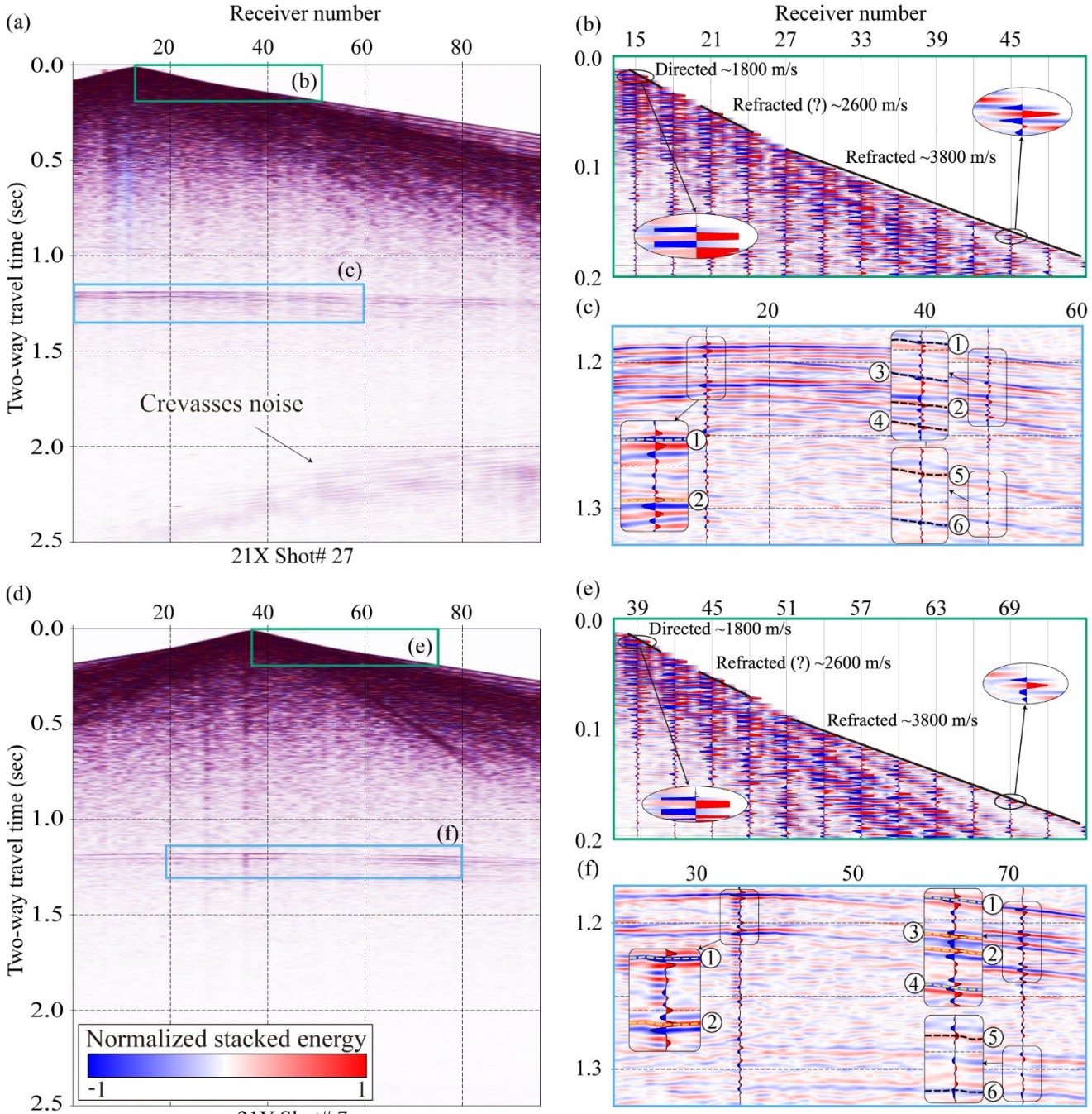

Figure 4: Raw shot records from seismic lines 21X (a) and 21Y (d). Panels (b) and (e) are zoomed-in views of the early arrival window (0.0–0.2 s) from panels (a) and (d), respectively, used to calculate the apparent velocities of the direct and refracted waves. These panels highlight that the first arrivals of both the direct wave (clipped for display) and the refracted wave exhibit positive polarity. The direct wave, propagating through the upper firn layer (0–25 m depth), shows an apparent velocity of approximately 1800 m s⁻¹, while the refracted wave in firn-ice transition has an apparent velocity of approximately 3800 m s⁻¹. Panels (c) and (f) are zoomed-

**in views of the deeper arrivals (1.1–1.4 s) from panels (a) and (d), respectively. Reflections from the ice–water (①) interface exhibit**
**negative polarity, whereas those from the water/sediment–bed (⑤) interface display positive polarity.**

## 3.2 Seismic data processing

Although seismic data acquired from glaciers share processing similarities with those of land-based surveys, glaciological
factors, such as surface cracks, crevasses, and strong winds, introduce substantial noise that can degrade data quality (Johansen
et al., 2011; Zechmann et al., 2018). Among these factors, linear noise generated by crevasses is particularly detrimental, often
obscuring key reflections (Dow et al., 2013). Hence, the glacier seismic data underwent multiple data processing sequences
focused on linear noise removal (Fig. 5). Acquisition geometry was added to the data using the raw data and geometry
information. Multiple data processing and noise removal processes were then carried out to increase the signal-to-noise ratio
(SNR).

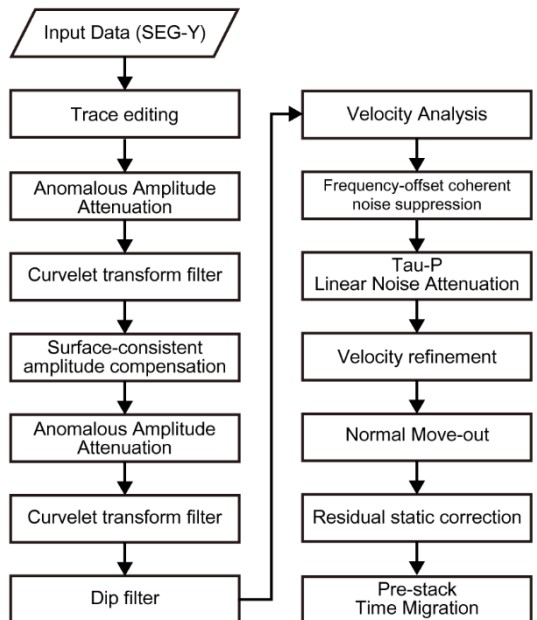

**Figure 5: Schematic of the seismic data processing workflow based on the Omega geophysical data processing platform (SLB),**
**including noise attenuation, amplitude correction, velocity analysis, and prestack time migration.**

The initial processing involved anomalous amplitude attenuation (AAA), implemented via a spatial median filter. This step
targets outlier amplitudes within a defined frequency band, attenuating anomalous signals through interpolation across
neighboring traces. A curvelet transform-based filter was subsequently applied to remove coherent noise. Curvelet
decomposition enables the separation of signals on the basis of dip angle and scale, allowing for the selective removal of
ground roll and other coherent noise components that differ in dip from true reflections (Oliveira et al., 2012). In this study,
linear coherent noise at later arrival times (>2.0 s) was effectively removed using this method.
Surface-consistent amplitude compensation (SCAR) and surface-consistent deconvolution were employed to normalize the
amplitude variability across shot gathers. These steps were followed by a second round of AAA and curvelet filtering to
suppress artifacts introduced during the compensation and deconvolution stages. Dip filtering was also applied to eliminate
spurious hyperbolic arrivals, which were manually identified and removed.
Velocity analysis was conducted at intervals of 40 common midpoints to construct a migration velocity model. Frequency–
offset coherent noise suppression (FXCNS) was used to attenuate linear-related noise, followed by Tau-p linear noise
attenuation (LNA), effectively reducing the noise associated with crevasse scattering. The final processing steps included
velocity model refinement, normal move-out (NMO) correction, and prestack time migration (PSTM). The specific parameters
employed for data processing, as well as the intermediate outcomes at each processing stage, are provided in the supplementary
information S2.
To increase imaging accuracy, a residual static correction was applied before migration using glacier surface elevation data.
The final migrated seismic section was produced using Kirchhoff PSTM. The migrated data have a center frequency of
approximately 180 Hz. Assuming seismic wave velocities between 1396 m s$^{-1}$ (water) and 3800 m s$^{-1}$ (ice), the corresponding
vertical resolutions, which are calculated using the quarter-wavelength criterion, range from approximately 2.0 m to 5.3 m.
The data can image both the top and bottom of a water column approximately 2 m thick or thicker.
**4 Seismic data processing results**
Figure 6 presents the PSTM results for the four seismic survey lines. On line 21X (Fig. 6a), a strong, laterally continuous
reflection with reverse polarity is observed at 0.3–4.8 km along the profile, and the two-way travel time (TWT) is
approximately 1.15–1.18 s. This reflection is interpreted as the glacier–lake interface (①). Approximately 25–30 ms below
this horizon, a normal polarity reflection (②) appears, likely representing a ghost signal associated with the primary glacier–
lake reflection. Between reflections ① and ②, a weak normal polarity reflection (③), presumed to represent an interface, is
observed. However, in some shot gathers, signal ③ appears with reverse polarity (Figure 4c), leading to partial cancellation
and ambiguity in layer interpretation. Approximately 25 ms later, an opposite polarity ghost reflection (④) follows. A deeper
normal polarity reflection is observed within 1.9–3.1 km at TWTs of 1.25–1.27 s (⑤), which is interpreted as the bed interface.
This is followed by a reverse polarity reflection 25–30 ms later (⑥), which is presumed to be the corresponding ghost of the
bed interface.

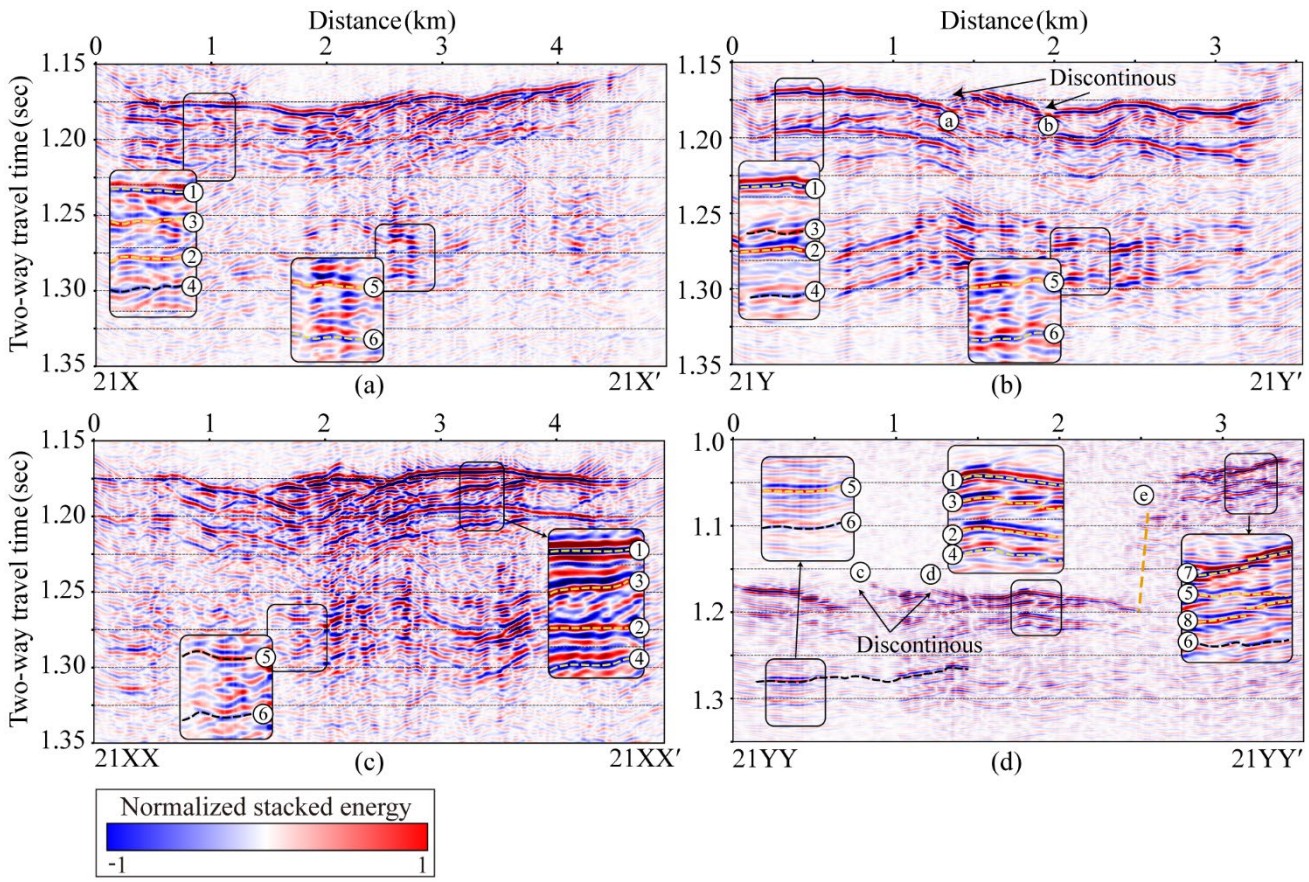

Figure 6: PSTM seismic sections for lines (a) 21X, (b) 21Y, (c) 21XX, and (d) 21YY prior to ghost removal. Ghost reflections appear 25–30 ms beneath the glacier–lake and lake–bed interfaces due to the 25 m source depth. See Table 2 for symbols definitions.

In line 21Y (Fig. 6b), similar features are observed. A reverse polarity reflection, interpreted as the glacier–lake interface (①), is observed within 0.1–3.2 km at TWT 1.17–1.18 s, with its ghost reflection (②), which exhibits normal polarity and appears 25–30 ms later. Between reflections ① and ②, a weak normal polarity reflection (③), presumed to represent an interface, is observed in some areas, followed approximately 25 ms later by an opposite polarity ghost reflection (④). A normal polarity reflection within 0.1–3.2 km at a TWT of 1.26–1.27 s is interpreted as the bed interface (⑤), followed by a reverse polarity ghost signal (⑥). Additionally, discontinuous reflections interpreted as subglacial scour-like features (SLF) are visible at approximately 1.3 km (ⓐ) and 1.9 km (ⓑ) along line 21Y at TWT 1.18 s (black arrows in Fig. 6b). These features may be associated with glacial erosion of the underlying substrate.

In line 21XX (Fig. 6c), a reverse polarity reflection, interpreted as the glacier–lake interface (①), is observed within 0–4.3 km at a TWT of 1.17–1.18 s. This reflection is followed 25–30 ms later by a normal polarity reflection (②), which is considered the ghost of the primary glacier–lake interface. Between reflections ① and ②, a weak normal polarity reflection

(③), presumed to represent an interface, is observed in some areas, followed approximately 25 ms later by an opposite polarity
ghost reflection (④). Further down the section, a normal polarity reflection (⑤) within 1.9–4.2 km at a TWT of 1.25–1.28 s
is interpreted as the bed interface, followed by its ghost reflection (⑥) 25–30 ms later.
On line 21YY (Fig. 6d), the glacier–lake interface (①) is marked by a strong, flat, reverse polarity reflection at 0–2.4 km and
a TWT of 1.17–1.20 s, followed by its normal polarity ghost (②) 25–30 ms below. A weak normal polarity reflection (③),
presumed to represent an interface, is observed between ① and ②, followed approximately 25 ms later by a opposite polarity
ghost reflection (④). Bed interface reflections (⑤) are observed within 0.2–2.4 km at TWTs of 1.27–1.29 s, followed by a
reverse polarity ghost (⑥) 25–30 ms later. Within 2.4–2.55 km and TWTs of 1.08–1.17 s, no coherent reflection is visible
due to the steeply dipping bed topography, as indicated by the dashed orange (ⓔ) line in Fig. 6d. Within 2.55–3.4 km and a
TWT of 1.03–1.09 s, a reverse polarity reflection (⑦), likely originating from a mildly dipping sedimentary surface, is
observed, followed by an opposite polarity ghost reflection (⑧). Additionally, although weak, reflection signal (⑤) and its
corresponding ghost (⑥) are also identified. Additionally, similar to observations on line 21Y, discontinuous reflections
interpreted as SLF surfaces appear at 0.7 km (ⓒ) and 1.2 km (ⓓ) along line 21YY at TWT 1.18 s (black arrows in Fig. 6d).
The discontinuous reflection signals identified on lines 21Y and 21YY are spatially aligned along the ice flow direction when
projected laterally (Fig. 3, dashed blue arrow). This alignment suggests that the observed discontinuities correspond to a
subglacial SLF surface formed by glacial motion. The SLF is visible predominantly on lines 21Y and 21YY, which are oriented
more perpendicularly to the ice flow direction, thereby enhancing the expression of lateral subglacial variability. In contrast,
lines 21X and 21XX are more parallel to the ice flow, resulting in a foreshortened view of the subglacial structures and a
relatively flat appearance in the seismic sections (Fig. 7).

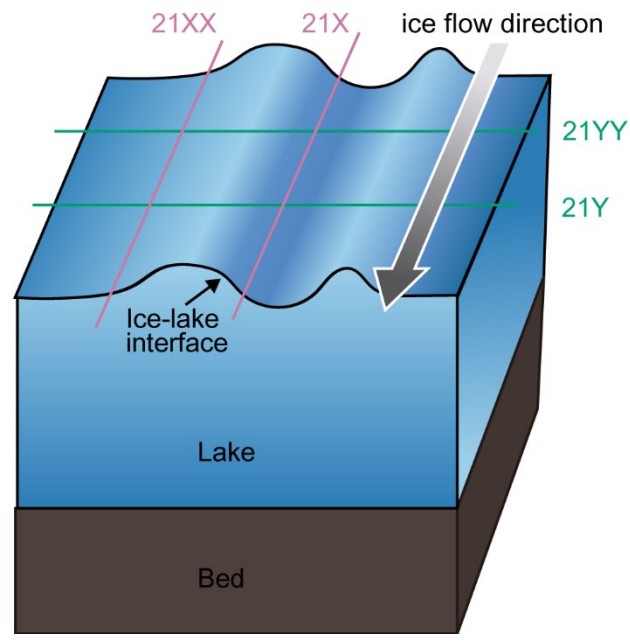


**Figure 7: Conceptual diagram illustrating the orientation of seismic survey lines relative to subglacial structures and the ice flow**
**direction, explaining the appearance of structural features in each line.**

## 5 Comparison between field data and synthetic seismograms

In all seismic profiles, the glacier–water interface (①) is characterized by strong, reverse polarity reflections. Following this,
a relatively weaker reflection (③) with limited lateral continuity, which may indicate an unconsolidated sediment layer, or an
unknown interface beyond the scope of current interpretation.
Interpreting field seismic data presents inherent challenges due to limited subsurface information, high levels of ambient noise,
and signal attenuation. These issues are particularly pronounced at the SLD2 site, where the absence of borehole data
introduces significant uncertainty and potential inaccuracies in depth estimations derived from PSTM sections. Such
limitations may lead to misinterpretations of stratigraphic boundaries (Herron, 2000; Yilmaz, 2001). To address these
challenges, this study developed a subsurface structural model and conducted a comparative analysis of synthetic seismograms
generated from the model with observed field data. Focusing on the interpretation of basal reflections beneath the subglacial
lake—excluding the glacier–lake interface (①)—two plausible structural models were proposed. Model 1 assumes the absence
of a sedimentary layer, in which reflection (③) is not present, and reflection (⑤) represents the base of the subglacial lake.
In contrast, Model 2 includes a sedimentary layer, where reflection (③) corresponds to the lake–sediment interface and
reflection (⑤) indicates the sediment–bedrock interface (Figure 8). The synthetic data were generated using a time-domain
forward modeling approach based on the staggered grid finite difference method (Graves et al., 1996). The velocity model
used in the simulation was constructed based on field velocity analysis and previously published data, and included
stratigraphic units representing firn, glacial ice, subglacial water, sediment, and bedrock. Each layer was assigned appropriate
P-wave velocities and density values. P-wave velocities in firn vary from 1525 to 3800 m s⁻¹ because density increases with
depth (Kirchner and Bentley, 1979; Picotti et al., 2015; Qin et al., 2024). Glacial ice has an average P-wave velocity of
approximately 3800 ± 5 m s⁻¹ at –2 ± 2 °C (Kohnen, 1974), while subglacial water has a velocity of approximately 1396 ± 2
m s⁻¹ at –1.75 ± 0.25 °C, with a salinity less than 1 PSU (practical salinity units) (Thoma et al., 2010; Tulaczyk et al., 2014).
The P-wave velocities of the sediment and bed were referenced from the Lake Vostok model value (Carcione & Gei, 2003).
Forward modeling was then conducted using the Ricker wavelet, with acquisition parameters matching those used in the field
survey (Table 3). We applied just the migration step in case of the synthetic dataset, as it is free of noise.

**Table 3: Parameters of the synthetic model.**

| Synthetic modeling parameters | | | |
|---|---|---|---|
| **Model size** | 3.5 km (distance) x 3 km (depth) | | |
| **Source** | Ricker wavelet (zero–phase), 60 Hz | | |
| | 25 m depth, 90-m interval | | |
| **Receiver** | 0 m depth, 15-m interval, 96 channel | | |
| **Grid spacing** | 0.5-m | | |
| **Sampling interval** | 0.1 ms | | |
| **Layer parameters** | Thickness (m) | Velocity (m s⁻¹) | Density (g cm⁻³) |
| Firn | 100 | 1525–3800 | 0.3–0.917 |
| Ice | 1887–2221 | 3800 | 0.917 |
| Water (Model 1, 2) | 53–82 / 10 | 1396 | 1.017 |
| Sediment (Model 2) | 120 | 2817 | 2.128 |
| Bed | | 5200 | 3.2 |


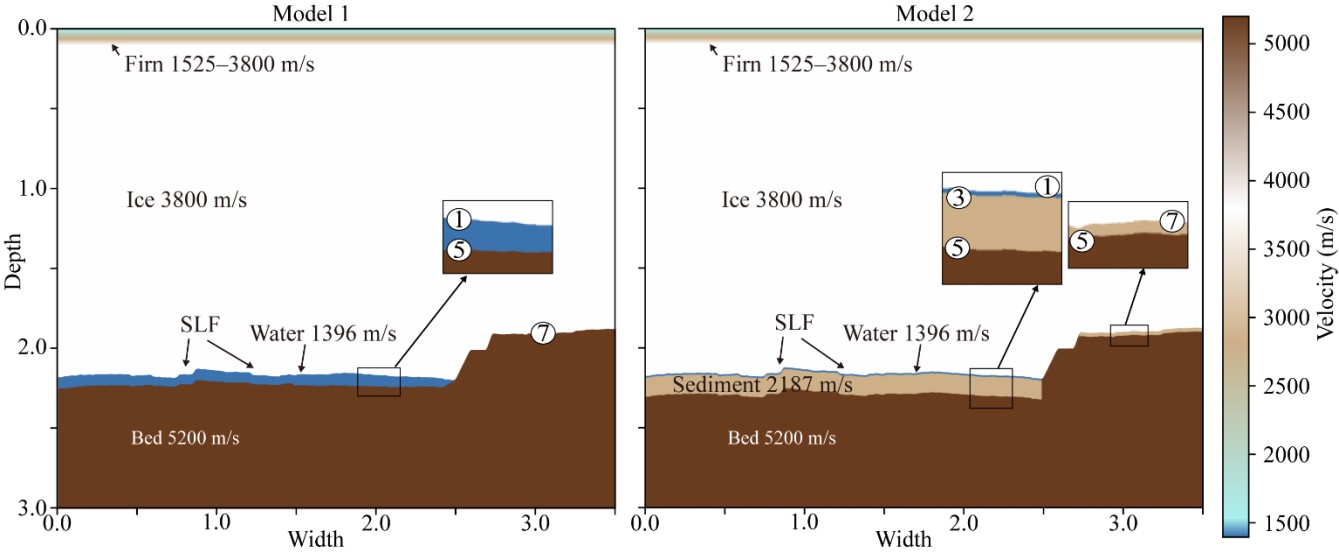


**Figure 8: P-wave velocity model used in forward modeling for line 21YY. The upper ~100 m represents firn with velocities ranging from 1525–3800 m s⁻¹ (Kirchner and Bentley, 1979; Picotti et al., 2015; Qin et al., 2024). The ice below this depth has a velocity of 3800 ± 5 m s⁻¹ (Kohnen, 1974), and the subglacial water layer has a velocity of 1396 ± 2 m s⁻¹ (Thoma et al., 2010; Tulaczyk et al., 2014). In Model 2, the velocity of 2817 m s⁻¹ for the sediment layer was taken from the lower sediment layer model of Lake Vostock (Carcione & Gei, 2003).**

Figure 9a compares the shot gather from seismic data line 21YY (left) with those from the synthetic datasets for Models 1 and 2 (center, right). A prominent reflection at a TWT of 1.17 s is observed in both datasets, corresponding to the glacier–lake interface (①). This reflection results in a high impedance contrast and reverse polarity due to the P-wave velocity difference between glacial ice and water. These features are consistent with previous observations at glacier–lake interfaces (Atre and Bentley, 1993; Brisbourne et al., 2023; Horgan et al., 2012; King et al., 2004; Peters et al., 2007; Woodward et al., 2010). A secondary reflection with normal polarity appears approximately 28 ms after the primary event (②) and is interpreted as a surface ghost reflection. This time delay corresponds to a seismic source depth of approximately 25 m, which is consistent with previous seismic analyses (Brisbourne et al., 2023; Schlegel et al., 2024). That is, assuming an average P-wave velocity of 1800 m s⁻¹ within the top 25 m, the TWT of the ghost reflection matches the expected delay:

$$\text{TWT}_{\text{ghost}} = \frac{2 \times 25 \text{ m}}{1800 \text{ m s}^{-1}} \approx 28 \text{ ms.} \tag{1}$$

Furthermore, considering that the acoustic impedance of air is approximately zero ($Z_{air} \approx 0$) and that of ice is $Z_{ice}$, the reflection coefficient ($RC$) for an upgoing wave at the air–ice interface can be approximated as follows:

$$RC = \frac{Z_{air} - Z_{ice}}{Z_{ice} + Z_{air}} \approx -1. \tag{2}$$

This implies that the polarity of the ghost reflection at the surface is reversed relative to the downgoing primary wave (Krail and Shin, 1990; Robinson and Treitel, 2008).

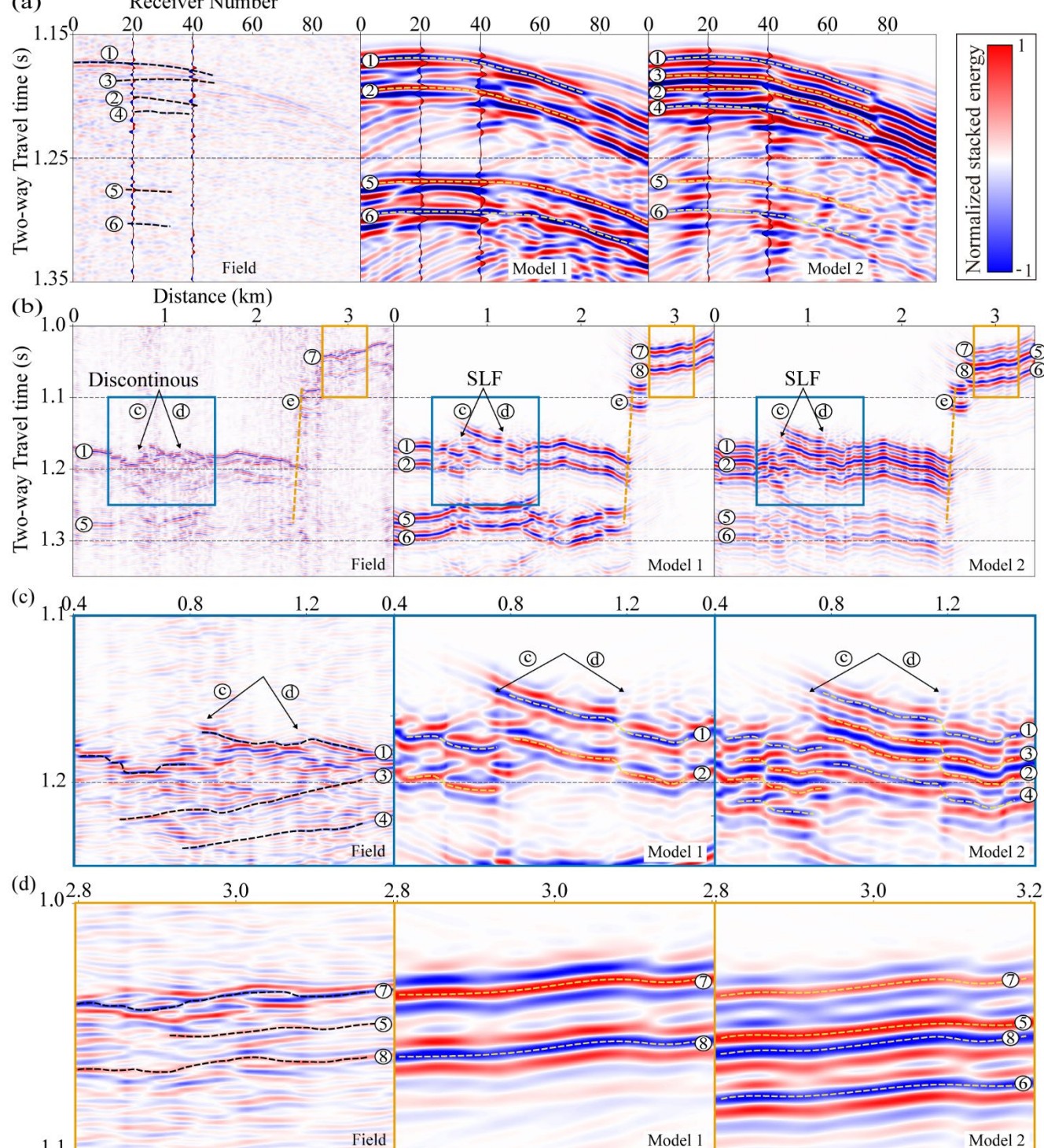

Figure 9: Comparison of field seismic data and synthetic results from Model 1 and 2. (a) Shot gathers at the same location from the 21YY field data (left) and synthetic models (center: Model 1, right: Model 2). (b) Comparison of PSTM images from the 21YY line

and the two synthetic models. (c) Enlarged views of discontinuous reflections. (d) Comparison of dipping bed reflections, showing shadow zones and steep basal topography.

Figure 9b compares the PSTM sections of the field data from line 21YY and the two synthetic models. Unlike the field data, the synthetic dataset is free from ambient noise and features a precise source–receiver geometry, resulting in clearer delineation of subsurface reflections and facilitating structural interpretation. In the PSTM sections of both the synthetic models (Model 1 and Model 2) and the field data, three primary reflection events (①, ⑤, ⑦) and their corresponding source-generated ghost reflections (②, ⑥, ⑧) are observed at similar two-way travel times. Reflections ①, ②, ⑤, and ⑥ also exhibit consistent polarity across the synthetic and field datasets. Additionally, the lateral discontinuities in reflections generated by the SLF structure implemented in the velocity model closely resemble those observed in the field data. The orange dashed line (ⓔ) delineates the shape of the bedrock forming the margin of the subglacial lake, interpreted to dip at approximately 52°.

Figure 9c presents an enlarged comparison between the field and synthetic PSTM sections, focusing on the region of lateral reflection discontinuities. In the field data, discontinuous reflections and associated low reflectivity observed at approximately 0.7 km and 1.2 km (TWT = 1.18 s) complicate interpretation. To simulate this feature, the velocity model incorporates a concave structure at the beneath the glacier, representing the SLF. The resulting reflection patterns in the synthetic section closely resemble those observed in the field data. In the field data, the reflection from the water–sediment interface (③) is weak and poorly defined, resulting in significant interpretational uncertainty. This is attributed in part to the diffusive interface of the unconsolidated upper sediment, which weakens reflection strength. Moreover, the short temporal separation between reflection ③ and the preceding ghost reflection (②) results in significant waveform interference, often producing a single, high-amplitude composite signal. Overlap between reflections ② and ③ leads to destructive interference, further complicating the identification of the interface. In some areas, the ghost reflection (④) is unaffected by such interference, allowing for an indirect estimation of the ③ interface using ghost travel-time differences. However, due to the difficulty in clearly resolving this interface throughout the dataset, Model 2 was constructed using a uniform geometry derived from the average time interval between reflections ① and ③, corresponding to a water depth of 10 m and a sediment thickness of 120 m. As a result, the arrival time and waveform characteristics of reflection ③ in the synthetic data exhibit slight discrepancies when compared to those observed in the field data.

Figure 9d presents a magnified comparison of regions synthetic and field to examine reflections from a dipping bed. Within 2.4–2.55 km and TWTs of 1.08–1.17 s, reflections are temporally dispersed, resulting in a shadow zone where coherent signals are absent. A noteworthy feature is the polarity reversal of reflections ⑦ and ⑧ between the field and synthetic datasets. In the velocity models, the ⑦ interface is defined as either the glacier–bedrock interface in Model 1 or the glacier–sediment interface in Model 2. In Model 1, the bedrock has significantly higher acoustic impedance than the overlying ice, due to its greater density and seismic velocity, resulting in a high amplitude reflection with normal polarity. Conversely, the sediment layer in Model 2 is assigned a lower seismic velocity but higher density relative to glacial ice, yielding a slightly higher impedance and thus a reflection of lower amplitude. However, the field data show that reflection ⑦ exhibits reversed polarity,

suggesting the presence of subglacial sediments with lower acoustic impedance than assumed in the models. This discrepancy may be explained by the presence of a dilatant till beneath the glacier, which can produce reverse polarity reflections depending on its physical properties. Booth et al. (2012) demonstrated that the seismic response of such tills is highly sensitive to variations in P-wave velocity, density, and thickness. In particular, their study showed that when the till forms a thin layer, reverse polarity reflections may occur. While the existence of such glacial sediments presents a plausible interpretation for the study area, the absence of reliable constraints on their seismic properties precluded their incorporation into the velocity models used in this study.

To further validate the interpretation, ice thickness estimates from the seismic data were compared with those derived from airborne IPR surveys (Ju et al., 2025) along four seismic lines (Fig. 10). Given the lack of spatial coincidence between seismic and IPR profiles, kriging-based two-dimensional interpolation (Isaaks and Srivastava, 1989) was applied to the IPR dataset to estimate the ice thickness at seismic line locations. The uncertainties associated with the IPR and seismic datasets are ±33.4 m and ±7.6 m, respectively at the 99% confidence level (Supplementary information S3). Consequently, the combined uncertainty of both datasets is approximately ±34 m. The root mean square error (RMSE) between the two datasets is calculated as approximately ±29 m. When excluding between 1.7 and 2.6 km along seismic line 21YY—within the light green shaded area in Fig. 10—where significant interpolation-induced smoothing effects occur in the IPR data, the RMSE is reduced to approximately ±25 m. This result indicates a high degree of consistency between the seismic and IPR datasets. The ice thickness derived from radar data is generally greater than that obtained from seismic data across most areas. This discrepancy may be attributed to an overestimation of the radar velocity. Ju et al. (2025) adopted a commonly used literature-based radar velocity of 0.169 m ns$^{-1}$, which may differ from the actual radar velocity in the study area. Additionally, the uncertainty in measuring the ice bottom in the radar data is ±32.7 m, and this must be considered when comparing the two datasets. Despite these factors, the two datasets show a high level of consistency within the uncertainty bounds. This consistency supports the mutual reliability of both methods and validates their integrated application for subglacial lake characterization.

As additional supporting evidence for this interpretation, a steeply dipping (ⓔ) bedrock interface observed along the 21YY line is consistently identified in both the seismic PSTM profile (Figure 9d) and the IPR-derived ice thickness graph (Figure 10), indicating a similar topographic transition in both datasets. This interface is interpreted as a structural margin delineating the lateral extent of SLD2 and likely functions as a hydrological barrier. The structural congruence observed in both seismic and radar data underscores the effectiveness of integrating these datasets to delineate the boundaries of subglacial lakes, particularly in regions characterized by complex basal topography.

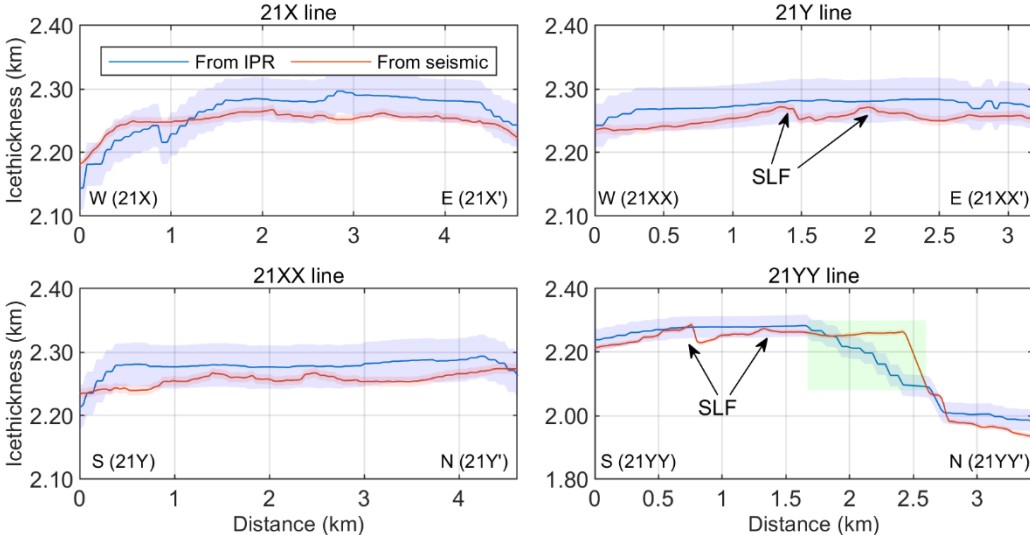


**Figure 10: A comparison of ice thickness estimates derived from seismic and kriging-interpolated IPR data (Ju et al., 2025) along**
**the four seismic survey lines reveals high overall consistency between the two datasets, despite localized discrepancies. The light**
**green shaded region in the 21YY line represents areas where interpolation contributes to the divergence between the two**
**measurement approaches. The light blue envelope represents the uncertainty bounds associated with the IPR-derived estimates,**
**while the light red envelope indicates uncertainty bounds for the seismic-derived estimates.**

## 6 Conclusion

Since 2016, the Korea Polar Research Institute (KOPRI) has conducted a series of geophysical investigations to study SLD2
(Subglacial Lake Cheongsuk) beneath David Glacier, beginning with airborne IPR surveys. In 2021, a seismic survey was
carried out to characterize the internal structure and water column of SLD2. The field seismic data revealed a strong, reverse
polarity reflection at the glacier–lake interface. In contrast, the basal reflections beneath the lake are less well-defined,
suggesting the presence of subglacial sediments. This ambiguity gives rise to two alternative interpretive scenarios based on
the presence or absence of a sedimentary layer.
Given this interpretational ambiguity regarding the sediment layer, two velocity models were constructed: Model 1, which
assumes the absence of sediment, and Model 2, which includes a sediment layer beneath the lake. Synthetic seismology was
generated using wave propagation modeling based on these models. Sediment thickness in Model 2 was uniformly assigned
using the average time difference calculated from selected areas of the dataset. Comparisons between the synthetic and field
PSTM sections show consistent TWT times and polarities for key reflection events at the glacier–lake interface, the lake–
bedrock interface in Model 1, and the sediment–bedrock interface in Model 2. Nevertheless, synthetic data generated by
modeling a velocity model that simplifies a complex geological structure has limitations in thoroughly explaining the entire
waveform of the complex field data. For example, subglacial sediments are generally expected to produce normal polarity
reflections due to acoustic impedance contrasts with overlying water. However, in field data, the polarity and clarity of the
water-sediment interface vary with the degree of sediment consolidation. In particular, the reverse polarity reflection observed
at the ice–sediment interface in the 21YY profile suggests the potential presence of dilatant till.
This study demonstrates the utility of seismic surveying for analyzing structural characteristics of subglacial lake environments
that are not identified with conventional radar. Furthermore, the integrated analysis of seismic and synthetic data provides a
quantitative structural model of the SLD2-A geometry beneath David Glacier. These results provide critical guidance for future
clean hot-water drilling. In particular, we identify an area within a 1 km radius of S 75.422°, E 155.441° as a suitable candidate
site, based on its broad spatial extent, minimum estimated water depth exceeding approximately 10 m, and absence of
contamination from surface field camps. Furthermore, we plan to conduct follow-up studies incorporating advanced processing
techniques such as deghosting, amplitude variation with offset analysis, and the development of a refined velocity model that
accounts for detailed firn-layer properties. These technical advancements are expected to enhance the resolution and precision
of seismic imaging and contribute to a deeper understanding of the subglacial environment.
**Data availability**
The ICESat-2 data used in this study are available from the National Snow and Ice Data Center (NSIDC). The seismic data
and ICESat-2 laser altimetry data used in this study are also available from the Korea Polar Data Center (KPDC) upon request
at https://dx.doi.org/doi:10.22663/KOPRI-KPDC-00001177. The maps related to Antarctica were created using the
Quantarctica dataset version 3.2 (Matsuoka et al., 2018).
**Author contributions**
HJ: Writing—original draft, investigation, methodology, conceptualization. SGK: Writing—original draft, methodology,
conceptualization, supervision. YC: Writing – original draft, data processing, modeling. SP: Data processing methodology.
MJL: Writing – original draft. HK: Hot-water drilling. KK: Investigation. YK: Investigation. JIL: Project administration,
Funding acquisition.
**Competing interests**
The authors declare that they have no known competing financial interests or personal relationships that could have appeared
to influence the work reported in this paper.
**Acknowledgments**
We express our sincere gratitude to Sungjun Jeon and the K-route team for their invaluable logistical support. We also extend
our appreciation to Do-youn Kwon, Jamin Park, Sanghyeok Seo, and Byeongguk Moon for their dedicated assistance in

seismic surveys. We name Subglacial Lake D2 Subglacial Lake Cheongsuk (SLC). The name Cheongsuk has a significant meaning, as it is the pen name of Dr. Yeadong Kim, the founder of the KOPRI and former president of the Scientific Committee on Antarctic Research (SCAR). Dr. Kim personally led the IPR and seismic surveys of Subglacial Lake Cheongsuk and coauthored this paper.

**Financial support**

This research was supported by KOPRI grants funded by the Ministry of Oceans and Fisheries (KOPRI project Nos. PE26080).

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
