# Peer review of "A seismic analysis of subglacial lake D2 (Subglacial Lake Cheongsuk)"

_EGUsphere, 2025_

## Author Comment (AC2)

We sincerely thank the Huw Horgan referee for their very helpful comments and for thoroughly reviewing the manuscript. The comments were very valuable and helpful in improving the clarity and quality of the manuscript. We have included all the comments and responded to them in detail below. Line numbers refer to the revised manuscript version.

**Comments from Huw Horgan (RC1)**

**Specific comments**:

The manuscript would benefit from the following:
1. Some additional justification of the survey location. Figure 3 shows that the survey location falls almost entirely outside of the active lake boundary from Smith et al., 2009. Please explain why this is the case. A useful addition to Figure 3 would be to show contours of equal hydropotential. This would further support the site selection, especially if they showed a hydropotential sink (closed contours of hydropotential.)

➔ We appreciate the reviewer's valuable comment regarding the clarification of site selection for the seismic survey. In response, we have added a reference to Ju et al. (2025), which provides a detailed description of the subglacial lake boundaries inferred from ice-penetrating radar (IPR) surveys. The location of the seismic survey lines in this study was carefully determined based on the lake extent proposed by Ju et al. (2025). To improve clarity, we have revised the manuscript to explicitly state that the seismic survey was conducted within the area where the presence of a subglacial lake was inferred from radar observations. Additionally, we updated Figure 3 to include the subglacial lake extent defined by Ju et al. (2025), along with a hydraulic head map, to provide more comprehensive spatial context for the survey design.

(Page 3, Lines 71-74) "Building upon these observations, Ju et al. (2025) subdivided the previously identified single subglacial water body at SLD2, as detected by ICESat altimetry, into three smaller subglacial lakes: SLD2-A, SLD2-B, and SLD2-C. Among these, SLD2-A represents the largest areal extent, and targeted seismic surveys were conducted over this area to obtain high-resolution information on the lake depth and basal structure."

[Figure]

**Figure 3: 21/22 seismic survey layout (black lines) overlaid on (a) bed elevation and (b) hydraulic head data from IPR results (Ju et al., 2025).**

2. Please include a more detailed description of the reasons for the unsuitability of the seismic data acquired previously. This would be of benefit to other researchers as it would allow them to avoid similar pitfalls.

➔ We have revised the manuscript to include a more detailed explanation of the technical issues and limitations encountered during the preliminary seismic campaign.

(Pages 5-6, Lines 125-137) "During the 2019/20 austral summer, a preliminary seismic survey was conducted over the SLD2-A region to evaluate the potential presence of a subglacial lake and to obtain initial information on its structural characteristics. Owing to limited field time and equipment constraints, the fold of coverage for all survey lines was restricted to 1, and all shot points were aligned near surface crevasses. Consequently, the acquired seismic data were significantly contaminated by strong linear coherent noise associated with crevasses, which severely degraded the signal quality of key reflectors, particularly reflections from the subglacial lake–bedrock interface. In addition, explosives are deployed within shallow boreholes (< 20 m depth), and owing to the absence of proper backfilling and the rapid timing of detonation, poor coupling between the explosives and the borehole walls further reduces energy transmission efficiency, resulting in overall low-quality reflection signals (Ju et al., 2024). As a result, due to the limitations of single-fold acquisition, stacking was not feasible, resulting in a low signal-to-noise ratio (SNR) and the presence of dominant coherent noise, rendering the seismic dataset unsuitable for quantitative structural interpretation. Nevertheless, the preliminary survey qualitatively confirmed the glacier thickness beneath SLD2-A and suggested the presence of subglacial water, providing critical baseline information that guided the methodology and survey design of the subsequent detailed seismic campaign conducted during the 2021/22 season."

3. Seismic data can be very hard to present. I think presentation could be improved here. Reflections from and ice over water interface are high amplitude and negative polarity. The follow issues occur to me:

    3.1 Figure 4. The image and zoom sections are too small for me to identify the dominant polarity in the basal returns. I suggest presenting exemplar shot records of ice over water and ice over rock, and including larger insets showing the basal returns.

➔ We have revised Figure 4 to improve the clarity of basal reflection polarity. Specifically, we have included example shot gathers for ice over water and ice over bedrock scenarios, each accompanied by enlarged inset panels that highlight the key reflection events and their respective polarities. Additionally, we have added an inset showing the direct and refracted wave velocities within the 0–0.2 s two-way travel time (TWT) range. This inset also includes an annotation of the direct wave signal to confirm the polarity of the source wavelet. These revisions provide a more comprehensive illustration of the reflection characteristics and improve interpretation accuracy.

(Page 8, Lines 160-169) "Figure 4 presents shot gather #27 from line 21X and shot gather #7 from line 21Y. In these shot gathers, the velocity of the direct wave is estimated to be approximately 1800 m/s, and the refracted wave velocity is approximately 3800 m/s. First-arrival analysis of the direct wave indicates a normal polarity, confirming the source waveform polarity. A prominent negative polarity reflection is observed at a two-way travel time (TWT) of approximately 1.2 s, interpreted as the glacier–lake interface. Approximately 25–30 ms later, a ghost reflection with normal polarity appears. A subsequent reflection at approximately 1.3 s TWT, showing normal polarity, is attributed to the lake–bed interface, followed by its negative polarity ghost reflection 25–30 ms later. In shot gather #27, noise originating from crevasses becomes apparent from approximately 2 s TWT. As the distance to the crevasses decreases, this noise increasingly overlaps with the primary reflection arrivals, complicating the interpretation."

[Figure]

**Figure 4: Raw shot records from seismic lines 21X (a) and 21Y (d). Panels (b) and (e) are zoomed-in views of the early arrival window (0.0–0.2 s) from panels (a) and (d), respectively, used to calculate the apparent velocities of the direct and refracted waves. These panels highlight that the first arrivals of both the direct wave (clipped for display) and the refracted wave exhibit positive polarity. The direct wave, propagating through the upper firn layer (0–25 m depth), shows an apparent velocity of approximately 1800 m/s, while the refracted wave traveling through glacier ice has an apparent velocity of about 3800 m/s. Panels (c) and (f) are zoomed-in views of the deeper arrivals (1.1–1.4 s) from panels (a) and (d), respectively. Reflections from the ice–water interface exhibit negative polarity, whereas those from the water–bed interface display positive polarity.**

3.2 Figure 6 images are too small as well. Making these subfigures larger would aid interpretation.

➜ We have revised Figure 6 by enlarging each subfigure of the PSTM (Pre-stack Time Migration) sections to enhance the visibility of key reflection signals. The time axis of the sections has been narrowed from the original two-way travel time (TWT) window of 1.0–1.5 seconds to a more focused window of 1.15–1.35 seconds, enabling a clearer visualization of the subglacial lake reflections.

[Figure]

**Figure 6: PSTM seismic sections for lines (a) 21X, (b) 21Y, (c) 21XX, and (d) 21YY prior to ghost removal. Ghost reflections appear 25–30 ms beneath the glacier–lake and lake–bed interfaces due to the 25 m source depth.**

3.3 Figure 9a looks to have an error. L257 states that this location represents ice over water but the dominant polarity of the basal return is +ve with small –ve side lobes. Also is the wiggle convention of +ve to the right being followed? Currently the right hand wiggle corresponds to the blue (-ve) color coding. This is confusing and should be corrected.

➔ We apologize for the mistake in the previous version of Figure 9, where the polarity was displayed incorrectly during the figure's production. We have revised Figure 9 accordingly. In the updated figure, negative polarity in the wiggle traces is now correctly shown on the left (blue) and positive polarity on the right (red). Additionally, the figure has been enlarged for better visibility, and the previously incorrect time axis labeling has been corrected. For clarity, the term "scour" was revised to "scour-like feature (SLF)" throughout the manuscript and figures.

[Figure]

**Figure 9: Comparison of synthetic and field seismic data. (a) Shot gather at the same location for synthetic (left) and 21YY field data (right). (b) PSTM comparison between the synthetic model and the 21YY line. (c) Enlarged views of discontinuous reflections (synthetic, field). (d) Comparison of dipping bed reflections (synthetic, field), showing shadow zones and steep basal topography.**

**Technical corrections**
4. The distance annotations shown in figures 2 and 6 should be shown on one of the basemaps.

➜ We revised Figures 1 and 2 to include the locations of SLD2-A, SLD2-B, and SLD2-C. Additionally, to help readers easily understand the orientation of the cross-sectional profiles in Figure 6, we added the X–X' line to Figure 3. Furthermore, the satellite altimetry in Figure 2 has been updated to include the most recent data through July 2024.

[Figure]

**Figure 1: Locations of subglacial lakes D1–D6 in the David Glacier region, Victoria Land, Antarctica (EPSG: 4326–WGS84).**

[Figure]

**Figure 2: Glacier surface elevation changes derived from ICESat-2 altimetry between 22 April 2019 and 12 July 2024. The X-axis corresponds to the 22 April 2019 dataset, and all subsequent elevation changes are referenced to this date. The light blue shaded region indicates the spatial overlap between the conventional SLD2 region identified by Smith et al. (2009) and our study region.**

5. The software used for processing the seismic data should be stated as the naming of routines is not always consistent across processing packages.

➔ We have revised the caption of Figure 5 for clarity. The original mention of "Omega geophysical data processing platform" has been updated to "Omega geophysical data processing platform (SLB)" to specify the software provider.

6. The distinction between active and inactive lakes should be made in the introduction.
➔ We have moved the definitions of stable and active subglacial lakes from Section 2.2 to the Introduction.

(Page 2, Lines 32-39) "Subglacial lakes in Antarctica are generally categorized as either stable or active. Approximately 80% of subglacial lakes in Antarctica are classified as stable subglacial lakes. These closed systems do not exhibit significant surface elevation changes and where subglacial water remains largely isolated, with minimal exchange due to slow and stable recharge and discharge cycles. The remaining 20% are classified as active subglacial lakes, which exhibit surface elevation changes due to episodic water drainage and refilling events (Livingstone et al., 2022). Such active lakes can reduce basal friction as they expand, thereby facilitating glacier flow and, in some cases, accelerating calving processes, ultimately influencing glacier dynamics (Bell et al., 2007; Stearns et al., 2008; Winsborrow et al., 2010)."

7. L79—83 This combination of data used to conclude that the region has contributed to SLR needs more rigour. As this is not the focus of the study I would instead suggest relying on an already published estimate. The ICESat2 surface elevation change results of Smith et al (2020) show the region upstream is thickening over the ICESat2 period.
➔ The relevant section of the manuscript has been revised based on the results and interpretations provided by Smith et al. (2020).

(Page 3, Lines 89-92) "According to Smith et al. (2020), satellite altimetry observations from ICESat-1 and ICESat-2 (2003–2019) indicate that the grounded portion of David Glacier experienced a mass gain of $3 \pm 2$ Gt yr$^{-1}$, whereas the adjacent ice shelves exhibited a mass loss of $-1.6 \pm 1$ Gt yr$^{-1}$. Although the overall mass balance of David Glacier currently appears stable, it remains uncertain how long this stability can be maintained."

8. L91 'with minimal exchange' I don't know if we know this. To my mind stable lakes just mean water is entering at the same rate it is exiting. More generally I would shift this description of active and stable lakes to the introduction.
➔ Please refer to our response to comment 6, where this issue is addressed in detail.

9. L109-110 repeat L61-63.
➔ We removed duplicate content from lines 109-110.

10. L112 'depressed basal elevations' Really it's the presence of hydropotential sinks as surface topography can dominate subglacial topography.
➔ The paper by Ju et al. (2025), which outlines the D2 subglacial lake area using radar data, has been recently published. We have cited this reference and revised several sentences in the manuscript to incorporate the updated information. Additionally, we have added the hydraulic head map to Figure 3 to provide more context for understanding the subglacial hydrology of the study area.

(Page 5, Lines 113-119) "To better constrain the lake's extent and basal conditions of SLD2, airborne IPR survey data from 2016/17 (Lindzey et al., 2020) and **2018/19 (Ju et al., 2025)** field campaigns indicate that glacier surface elevations in the SLD2 region range from approximately 1820 to 1940 m, with ice thicknesses varying between 1685 and 2293 m. Furthermore, the observations of moderately enhanced radar bed echoes relative to the surrounding area, elevated specularity values (>0.4), depressed basal elevations (≤–350 m), **the presence of a Bain-like topography, a lower hydraulic head than the surroundings,** and low hydraulic gradients (≤ 0.84°) collectively suggest a high potential for the presence of subglacial water beneath SLD2. (**Ju et al., 2025**; Lindzey et al., 2020)."

11. L123 'deployed'-> acquired

➔ (Page 6, Line 139) We have changed from
"A total of four seismic lines were **deployed** and designated 21X, 21Y, 21XX, and 21YY" to
"A total of four seismic lines were **acquired** and designated 21X, 21Y, 21XX, and 21YY".

12. Figure 3. Add hydropotential contours.

➔ Please refer to our response to comment 1, where this issue is addressed in detail.

13. Figure 4. Consider displaying fewer shots and making them larger so polarity can be more easily identified.

➔ Please refer to our response to comment 3.1, where this issue is addressed in detail.

14. L158 'A geometry setup was performed...' -> Acquisition geometry was added to the data...

➔ (Page 10, Line 184) We have revised from
"**A geometry setup was performed** using the raw data and geometry information" to
"**Acquisition geometry was added to the data** using the raw data and geometry information"

15. L159 or soon after – state what software was used for processing.

➔ Please refer to our response to comment 5, where this issue is addressed in detail.

16. Figure 5. Consider showing shot record and zoom before and after processing.

➔ We have added the shot records and PSTM results before and after processing in the Supplementary Information.

17. L181-182. Please state what you are reporting for resolution. (Looks like ¼ wavelength at for ice velocity at the upper end)

➔ We have revised the manuscript to explicitly state that the vertical resolution was calculated based on the quarter-wavelength criterion. The resolution range of 2.01 to 5.27 m was derived using the central frequency of 180 Hz and the range of seismic velocities (1395–3800 m/s) obtained in this study.

(Page 11, Lines 210-212) "Assuming seismic wave velocities between 1395 m/s and 3800 m/s, the corresponding vertical resolutions, **which are calculated using the quarter-wavelength criterion**, range from approximately 2.01 m to 5.27 m. This resolution is adequate for imaging SLD2."

18. Figure 6. These are too small for me to examine polarity. Please increase in size. You shouldn't need to reproduce the basemap here if it in presented well previously.

➔ Please refer to our response to comment 3.2, where this issue is addressed in detail.

19. L201 'These features may be associated with glacial erosion....'

➔ It is unclear what the question is intended to comment on.

20. L240 'P-wave velocity...is faster...' Strictly speaking it's an impedance increase.

➔ We have revised the sentence for clarity.

(Page 14, Lines 269-271) "Additionally, on line 21YY, the reflection polarity at the ice–bedrock interface appears as normal polarity, **which indicates an increase in the acoustic impedance**. In other words, this suggests that the P-wave velocity of the bedrock is **higher** than that of the overlying ice."

21. Figure 9. There are some issues with polarity discussed above. There looks to be a polarity reversal up the step in Fig 9b, which would be compelling and a nice example of how seismic data can show abrupt changes in water at the bed but again it's hard to see in the field data.

➔ Please refer to our response to comment 3.3, where this issue is addressed in detail.

22. L309-310 'hydrological barrier' hard to say without knowing surface. Again hydropotential contours would be helpful here.

➔ Please refer to our response to comment 1, where this issue is addressed in detail.

23. The conclusion could include statements on the mismatch between the active lake boundary and the area surveyed here and could suggest a location for direct access.

➔ Please refer to our response to comment 1, where this issue is explained in detail. Additionally, we have included a specific recommendation for the potential direct access drilling site in the conclusion section to support future subglacial exploration efforts.

(Page 19, Lines 367-370) "Ultimately, this study identifies the area within a 1 km radius of S 75.422°, W 155.441° as a suitable candidate site for clean hot-water drilling, given its wide spatial extent, minimum estimated water depth exceeding approximately 50 m, and absence of contamination from surface field camps. The site is therefore considered highly appropriate for future exploration of active subglacial lakes."

**Reference:**

Ju, H., Kang, S., Han, H., Beem, L. H., Ng, G., Chan, K., Kim, T., Lee, J., Lee, J., Kim, Y., and Pyun, S.: Airborne and Spaceborne Mapping and Analysis of the Subglacial Lake D2 in David Glacier, Terra Nova Bay, Antarctica, J. Geophys. Res.: Earth Surf., 130, https://doi.org/10.1029/2024jf008142, 2025.

Smith, B. E., Fricker, H. A., Gardner, A. S., Medley, B., Nilsson, J., Paolo, F. S., Holschuh, N., Adusumilli, S., Brunt, K., Csatho, B., Harbeck, K., Markus, T., Neumann, T., Siegfried, M. R., and Zwally, H. J.: Pervasive ice sheet mass loss reflects competing ocean and atmosphere processes, Science, 368, 1239–1242, https://doi.org/10.1126/science.aaz5845, 2020.

*Supplement of*

**Seismic data analysis for subglacial lake D2 beneath David Glacier, Antarctica**

Hyeontae Ju, Seung-Goo Kang, et al.

*Correspondence to:* Seung-Goo Kang (ksg9322@kopri.re.kr)

The copyright of individual parts of the supplement might differ from the article license.

S1. Seismic data processed parameters and results

This study utilized the Omega geophysical data processing platform (SLB) for seismic data processing. Among the various processing steps, we provide below the key parameters applied during procedures that directly influence the ice–bedrock interface signal, such as noise attenuation.

1.   Anomalous amplitude attenuation (AAA) for the 1$^{st}$ round

AAA is a frequency-domain filtering technique designed to suppress spatially coherent anomalous amplitudes such as swell noise and rig noise, by comparing amplitude spectra across traces and attenuating outliers based on spatial median statistics.

The method identifies frequency bands with anomalous energy by comparing each trace's amplitude spectrum within a spatial window to the median of its neighboring traces. Detected anomalies are either scaled or replaced using interpolated values from adjacent traces, preserving relative amplitude relationships. Key parameters include TIME, which defines the temporal window of threshold application; THRESHOLD FACTOR, which sets the amplitude level considered anomalous; and

SPATIAL MEDIAN WIDTH, which specifies the number of adjacent traces used for median computation. Proper tuning of these parameters is essential to avoid signal distortion while effectively attenuating coherent noise. AAA is particularly useful in prestack data conditioning as it enhances seismic data quality without compromising true subsurface reflections (SLB,

2025a).

●    SPATIAL MEDIAN WIDTH: 21 traces

●    Threshold factor tables:

| TIME | THRESHOLD FACTOR |
| --- | --- |
| 0 | 15 |
| 1000 | 10 |
| 3000 | 7 |
| 4000 | 6 |

2.   Curvelet transform-based filter for 1$^{st}$ round (Figure S43b)

Curvelet Transform is a multi-scale, multi-directional decomposition technique that provides a sparse representation of seismic data by capturing curved wavefronts more efficiently than conventional fourier or wavelet transforms. An important aspect of the Curvelet Transform implementation involves user-defined control over the scale and angle bounds that determine which components of the data will be transformed. The LOWER BOUND OF SCALE and HIGHER BOUND OF SCALE specify the range of spatial frequencies (scales) to be included in the transform. Lower scales correspond to coarse, low-frequency components, while higher scales capture fine, high-frequency structural details. The LOWER BOUND OF ANGLE and

HIGHER BOUND OF ANGLE define the directional sectors (angles) within each scale to be analyzed. This allows selective enhancement or suppression of events based on their dip or propagation direction (SLB, 2025b). Figure S1 illustrates how the f–k domain is partitioned into curvelet panels by scale and angle. Adjusting these bounds allows for targeted signal processing, such as isolating curved events or attenuating directionally coherent noise. These parameters provide valuable flexibility in customizing the transform for specific seismic applications.

●   Panel manager

| LOWER BOUND OF THE SCALE | HIGHER BOUND OF THE SCALE | LOWER BOUND OF THE ANGLE | HIGHER BOUND OF THE ANGLE |
|---|---|---|---|
| 2 | 2 | 1 | 3 |
| 2 | 2 | 8 | 10 |
| 3 | 3 | 1 | 6 |
| 3 | 3 | 13 | 18 |
| 4 | 4 | 1 | 6 |
| 4 | 4 | 13 | 18 |

[Figure]

**Figure S1: Illustration of the panel manager. In the f-k domain, the hatched area is identified as noise and removed accordingly.**

3.   Surface-consistent deconvolution

Surface-Consistent Deconvolution is a technique for generating and applying deconvolution operators that are consistent across seismic sources, receivers, offset ranges, and CMP locations (SLB, 2025c; Yilmaz, 2001).

Key processing parameters used in this workflow include:

●   CONSTANT_ACOR_LENGTH = 100: Defines the half-length of the autocorrelation window used in operator

 design, balancing spectral resolution and filter stability.

● WHITE NOISE PERCENT = 0.01: Adds 1% white noise to stabilize the autocorrelation estimation and prevent
over-whitening of the signal.
● PREDICTION DISTANCE = 2.5: Specifies the prediction lag in the predictive filter design; this parameter
controls the temporal range of the filter's effect, influencing multiple suppression and resolution.

4. Anomalous amplitude attenuation (AAA) for the 2nd round
● Spatial median width: 11 traces
● Threshold factor tables:

| Time | Threshold factor |
|------|------------------|
| 0 | 8 |
| 1000 | 6 |
| 3000 | 4 |
| 4000 | 3 |

5. Curvelet transform-based filter for the 1st round: same as 1st round parameter

6. Frequency-offset coherent noise suppression (Figure S3.c)
The frequency–offset (F-X) Coherent Noise Suppression (FXCNS) module is designed to attenuate near-surface shot-
generated coherent noise, such as dispersive surface waves and trapped modes, which interfere with primary seismic reflections,
particularly in 3D shot or receiver gathers with irregular spatial sampling (Hildebrand, 1982). FXCNS operates in the frequency
domain by modeling coherent noise using fan filters and estimating it in a least-squares sense for each trace based on local
neighbors within a specified azimuthal sector. The estimated coherent noise is then subtracted from the original signal,
preserving true reflection events (SLB, 2025d).
● LOW PASS VELOCITY: 100
● LOW STOP VELOCITY: 300
● HIGH PASS VELOCITY: 8000
● HIGH STOP VELOCITY: 10000

[Figure]

     (a) Raw data shot gather of synthetic            (b) After pre-stack time migration

**Figure S2: Results before and after data processing. (a) Synthetic raw data of shot gather #1. (b) Result after pre-stack time**
**migration.**

[Figure]

**Figure S3: Results at each stage of data processing. (a) Shot gather #1 from 21YY. (b) Removal of high-frequency random noise and**
**coherent linear noise. (c) Application of a frequency-offset coherent noise filter and tau-p linear noise attenuation for surface wave**
**removal. (d) Result after applying pre-stack time migration.**

**References**

Hildebrand, S. T.: Two representations of the fan filter, Geophysics, 47, 957–959, https://doi.org/10.1190/1.1441363, 1982.

SLB: *Omega Geophysical Processing System – Anomalous Amplitude Noise Attenuation (ANOMALOUS_AMP_ATTEN) Module Documentation*, Version 27.14, SLB manual, Houston, TX, 2025a.

SLB: *Omega Geophysical Processing System – Curvelet Transform (CURVELET_TRANSFORM) Module Documentation*, Version 22.1, SLB manual, Houston, TX, 2025b.

SLB: *Omega Geophysical Processing System – Surface-Consistent Deconvolution Analysis (SC_DCN_SPCTRL_ANL) Module Documentation*, Version 13.18, SLB manual, Houston, TX, 2025c.

SLB: *Omega Geophysical Processing System – F-X Coherent Noise Suppression (FXCNS) Module Documentation*, Version 3.16, SLB manual, Houston, TX, 2025d.

Yilmaz, Ö.: Seismic data analysis: Processing, Inversion, and Interpretation of Seismic Data, Appendix B.8 (Surface-Consistent Deconvolution), Society of Exploration Geophysicists, 262–266 pp., 2001.

---

## Author Comment (AC3)

We sincerely thank the anonymous referee for their very helpful comments and for thoroughly reviewing the manuscript. The comments were very valuable and helpful in improving the clarity and quality of the manuscript. We have included all the comments and responded to them in detail below. Line numbers refer to the revised manuscript version.

**Comment from anonymous reviewer (RC2)**

**Specific Comments:**

1. I wonder if the extensive processing scheme is fully necessary, given reflections are quite strong in the raw data. For example, two rounds of curvelet filtering and spatial filtering via interpolation seem redundant. At the very least, could you provide more information about some of the key filtering parameters? Specifically, to remove ground roll?

➔ As you noted, the glacier-lake reflections in the raw radar data indeed exhibit high amplitude. Consequently, our data processing strategy prioritizes the preservation of these strong reflection signals while selectively attenuating various forms of noise—including random noise, crevasse-related noise, and ground roll—that may obscure meaningful subsurface features. The two rounds of curvelet-based filtering and spatial filtering via median interpolation (Anomalous Amplitude Attenuation (AAA) based on the Omega geophysical data processing platform from SLB) were carefully designed with distinct objectives at different stages of the workflow:

First Stage (Initial Filtering of Raw Data):
The initial AAA and curvelet filtering were applied to suppress high-frequency random noise and coherent linear noise that becomes prominent at later two-way travel times (beyond 2 seconds). These noise components are particularly detrimental to the identification of deep or weak reflectors, and their suppression enhances overall signal-to-noise ratio (SNR) without degrading the primary lake-related reflections.

Second Stage (Post-Processing Filtering):
Following surface-consistent deconvolution and gain corrections, residual noise structures—particularly remnants of ground roll and shallow-layer reverberations—persist in the dataset. The second round of curvelet filtering and AAA was implemented to target these residuals, improving the lateral continuity of true reflectors while further mitigating spatially coherent noise that could lead to misinterpretation.

To support transparency and reproducibility, we have included the processed field data and the parameters used at each major step in the revised Supplementary Materials. These examples demonstrate the efficacy of the adopted filtering scheme in retaining high-amplitude lake reflections while effectively eliminating noise.

2. In figures, be a bit more specific about geometry. Add a 21X' on one end for example to see which direction the record sections are. You can figure it out, but it might make them a bit clearer

➔ We have added a reference to Ju et al. (2025), which provides a detailed description of the subglacial lake boundaries inferred from ice-penetrating radar (IPR) surveys. The location of the seismic survey lines in this study was carefully determined based on the lake extent proposed by Ju et al. (2025). We updated Figure 3 to include the subglacial lake extent defined by Ju et al. (2025), along with a hydraulic head map, to provide more comprehensive spatial context for the survey design. Additionally, to help readers easily understand the orientation of the cross-sectional profiles in Figure 6, we added the X–X' line to Figure 3.

[Figure]

**Figure 3: Seismic survey layout (black lines) overlaid on (a) bed elevation and (b) hydraulic head data from IPR results (Ju et al., 2025).**

3. Figure 4 and 6: the colorbar is not appropriate. My understanding of what's plotted is the normalized seismic section between -1 and 1, which would be equal (or rather proportional, equal after accounting for geometric spreading, source, attenuation etc) to the reflection coefficient only at normal incidence. I think the colorbar should either be labelled as normalized stacked energy or actual units given.

➜ We have revised Figures 4 and 6 by changing the colorbar label from "Reflection coefficient" to "Normalized stacked energy" to represent the displayed seismic attributes more accurately.

4. On this point, I wonder why not try to calculate an absolute reflectivity curve (AVO Horgan et al. 2021; Peters et al. 2008) or relative reflectivity curve, to confirm you are truly seeing water at the base rather than some kind of wet sediment layer, which would also result in a negative polarity PP reflection. Even if you just do this for one of the lines, the coherence with the analytical reflectivity curve would make the argument much stronger.

➜ We sincerely appreciate the reviewer's valuable suggestion. We conducted seismic surveys in areas with a high likelihood of basal water presence, as indicated by radar observations, based on the recently published study by Ju et al. (2025). We fully acknowledge that AVO analysis is a powerful tool for distinguishing between basal water and wet sediments. However, AVO analysis typically requires a sufficiently wide range of incidence angles to derive reliable reflectivity curves. We think that conventional AVO analysis would be difficult to apply due to the limited offset range resulting from the short length of the seismic line.

We fully agree with the scientific merit of the approach suggested by the reviewer. In response, and as part of our ongoing efforts to enhance subsurface interpretation, we are planning further research aimed at implementing deghosting and AVO analysis, as well as developing a refined velocity model that incorporates detailed firn-layer velocity structures. These improvements are expected to facilitate more advanced seismic processing, higher-resolution imaging of subglacial lake structures, and ultimately a more accurate understanding of the basal environment in future investigations.

We have added content to the conclusion.
(Page 19, Line 370-373) "Furthermore, we plan to conduct follow-up studies incorporating advanced processing techniques such as deghosting, amplitude variation with offset (AVO) analysis, and the development of a refined velocity model that accounts for detailed firn-layer properties. These technical advancements are expected to enhance the resolution and precision of seismic imaging and contribute to a deeper understanding of the subglacial environment in future investigations."

5. Figure 6e: I'm a bit confused on the interpretation here. The glacier-bed ghost (4) appears to be normal polarity, while the direct reflection appears to be reverse polarity? This seems opposite to the discussion in the analysis. I'm wondering if I am seeing something wrong, or perhaps the color scale is flipped for this inset?

➔ We have enlarged Figure 6 to enhance clarity and make key features more visible.

[Figure]

**Figure 6: PSTM seismic sections for lines (a) 21X, (b) 21Y, (c) 21XX, and (d) 21YY prior to ghost removal. Ghost reflections appear 25–30 ms beneath the glacier–lake and lake–bed interfaces due to the 25 m source depth.**

6. Scours: Really interesting interpretation, but I wonder how you can be sure these features are from depths rather than near surface or englacial crevassing? They look similar to observations from stations near shallow faults which show significant scattering or spurious arrivals from interactions with reflected phases. I also wonder if you could use cross correlation/autocorrelation with the direct

P to increase the coherence of the reflected arrivals and to better resolve the shape of these features? Further, while I see that the scour features interfere significantly in the synthetic, the observed amplitude is still comparable to the direct reflections, which is again not true in the data. I wonder if instead, the near surface scattering argument is invoked, would you expect significant defocusing and thus lower amplitude?

➔ We appreciate the reviewer's insightful comments. For clarity, the term "scour" was revised to "scour-like feature (SLF)" throughout the manuscript and figures.

The SLF structure is clearly observed along the Y-line but is not readily identifiable along the X-line. In particular, some end-shot records from the X-line exhibit overlapping signals between the ice–bedrock interface reflection and surface waves or crevasse-induced noise, which hinders interpretation. However, in the Y-line records, such scattering signals predominantly appear after TWT 1.5 seconds and are therefore clearly distinguishable from the depth range where the SLF structure is observed. Moreover, in the synthetic dataset, surface wave and crevasse-related noise do not overlap with the ice–bedrock interface reflection signals. Consequently, it is unlikely that the SLF signal could be misinterpreted as surface scattering or surface wave noise. Furthermore, the surface topography of the study area is very flat, and the shallow sediment layer exhibits a similar spatial distribution. Therefore, the likelihood of surface or shallow structures generating scattering signals that could be mistaken for SLF is very low. Additionally, no reflection patterns or polarity characteristics typical of SLF structures are observed in the shallow parts of the seismic profiles.

The synthetic velocity model is discretized, leading to diffraction signals around the SLF structure in the synthetic seismograms. These diffractions interfere with nearby reflectors during migration. As the reviewer noted, it is still possible to interpret these features as defocusing effects caused by near-surface scattering. However, given the geological and geophysical conditions of the site, we believe it is more likely that the SLF structure is located at the bottom of the glacier.

Nonetheless, to fully exclude the possibility of near-surface scattering or spurious arrivals from shallow fault zones, further verification is necessary. We agree with the reviewer that applying cross-correlation and autocorrelation techniques will strengthen the connection with direct P-waves and enable additional validation of reflection consistency. We are currently performing follow-up research using these methods, along with improved velocity modeling, to improve resolution and achieve greater accuracy in deep structural interpretation.

7. I am wondering about the firn velocity model used: If I do a linear interpolation, I see that the average firn velocity in the upper 25 meters is around 1800 m/s, consistent with what is stated in the text. However, a consistent feature of the firn is the steep velocity gradient/density near the surface which shallows with depth, which is very nonlinear. I believe this effect is probably small, but in terms of the raytracing, the incident angles of the incoming waves in the simulations of a linear vs nonlinear firn will be very different, which affects reflectivity. If you calculate your own Vs firn model, or if you use an empirical relation/calibration for your firn model, you could solve for a firn to ice transition depth which would be another interesting finding of the paper.

➔ We appreciate the reviewer's insightful comment regarding the firn velocity structure. As pointed out, firn velocity generally increases non-linearly with depth due to rapid density and velocity changes near the surface (Agnew et al., 2023; Picotti et al., 2024). However, accurately developing a site-specific firn velocity model remains very challenging in practice, since firn properties can vary greatly depending on factors like local compaction rates, the ratio of fresh to old snow, moisture content, and seasonal temperature changes.

In the study area above the D2 subglacial lake, field observations confirmed that the snow and firn surface is highly compacted. During fieldwork, the surface was so solid that it did not break under foot traffic. Additionally, when excavating about 5–10 cm to install receivers, we observed that the near-surface snow layer had a high density, similar to solid ice.

Quantitatively, the first-arrival travel times recorded by receivers directly above the shot points (offset = 0 m) ranged from approximately 10.6 to 15.5 ms. Considering the estimated shot depth of approximately 25 m, the apparent average velocity for the top 25 m is calculated to be approximately m/s (Figure R1).

[Figure]

Figure R1: (Left) First-arrival travel times recorded at offset = 0 m along survey line 21X. (Right) Apparent velocity for the upper 0–25 m depth estimated based on the measured first-arrival times.

Additionally, the apparent velocity in the 0–25 m depth range, estimated from the ghost signal with an approximate delay of 28 ms, is about 1785 m/s. Velocity analysis of the shot records shows a three-layer structure: the direct wave velocity in the top layer (0–50 m depth) is roughly 1800 m/s; in the middle layer (50–100 m), the direct wave velocity increases to around 2600 m/s; and the refracted wave, related to glacial ice, has a velocity of approximately 3800 m/s (Figure R2). Accordingly, we revised Figure 4 to display the velocities of both direct and refracted wave, reflecting these considerations.

[Figure]

Figure R2: Direct and refracted wave velocities estimated from End-shot data (Shot record #9) along survey line 21X.

[Figure]

**Figure 4: Raw shot records from seismic lines 21X (a) and 21Y (d). Panels (b) and (e) are zoomed-in views of the early arrival window (0.0–0.2 s) from panels (a) and (d), respectively, used to calculate the apparent velocities of the direct and refracted waves. These panels highlight that the first arrivals of both the direct wave (clipped for display) and the refracted wave exhibit positive polarity. The direct wave, propagating through the upper firn layer (0–25 m depth), shows an apparent velocity of approximately 1800 m/s, while the refracted wave traveling through glacier ice has an apparent velocity of about 3800 m/s. Panels (c) and (f) are zoomed-in views of the deeper arrivals (1.1–1.4 s) from panels (a) and (d), respectively. Reflections from the ice–water interface exhibit negative polarity, whereas those from the water–bed interface display positive polarity.**

Based on the field observations, we applied a 1D velocity model assuming a linear increase in velocity from the surface down to 100 m depth, using an apparent velocity of approximately 1800 m/s for the upper 0–25 m layer. Although we attempted to apply nonlinear firn velocity models as suggested by the reviewer (e.g., Yang et al., 2024; Picotti, Carcione, & Pavan, 2024; Agnew et al., 2023), we found that, given the observed first-arrival times in this region, these models either yielded physically unrealistic results or led to numerical instabilities.

Nevertheless, we fully acknowledge the importance of more accurately representing the firn velocity structure. To this end, we are currently conducting additional data analysis and high-resolution velocity inversion, with the goal of applying more advanced firn velocity models such as those recently proposed by Agnew et al. (2023) and Picotti et al. (2024). These improvements will be incorporated into a future study to enhance the robustness and physical realism of the velocity model.

We have added content to the conclusion (Page 19, Line 371-374).

8. I wonder if the full processing scheme need be applied to the synthetic data. Given the extensive processing combined with the lack of ambient noise/crevasse scattering, the data should basically be clean enough to examine on its own. I understand for consistency's sake why the same processing is applied, but here, I would emphasize that the synthetics before and after processing remain pretty much the same. If they do not, that could point to artifacts being introduced in your processing scheme.

➜ As correctly pointed out, excessive or unnecessary processing steps may introduce artifacts, particularly in synthetic datasets that lack ambient noise and crevasse scattering. In response, we have revised our approach and now apply only the migration step to the synthetic data. We agree that this modification is more appropriate and better aligned with the purpose of using synthetic data. Accordingly, the main text has been updated to reflect this change. In addition, we have included both pre- and post-migration results of the synthetic dataset in the revised Supplementary Materials, allowing for transparent evaluation of any potential processing-related artifacts.
(Page 14, Line 274) We have revised from
"The same seismic processing sequence applied to the field data (Section 3.2, Fig. 5) was subsequently applied to the synthetic dataset to produce a PSTM image for comparison" to
"We applied just the migration step in the case of the synthetic dataset, as it is free of noise".

9. I think Figure 9a also has a polarity issue. This is showing the synthetic and observed seismic data shot gathers with arrivals corresponding to the glacier lake interface, but the first arrival in both synthetic and observed appear to be normal polarity as opposed to the reverse polarity that is stated. Am I interpreting these figures wrong? IN addition, back to the glacier-bed interface, the synthetic phases 3, 4 representing the glacier-bed and ghost have normal and reverse polarity respectively, while the field observations seem to have reverse and normal polarity respectively. Is this a color bar issue, or am I interpreting these images wrong?

➜ We apologize for the mistake in the previous version of Figure 9, where the polarity was displayed incorrectly during the figure's production. We have revised Figure 9 accordingly. In the updated figure, negative polarity in the wiggle traces is now correctly shown on the left (blue) and positive polarity on the right (red). Additionally, the figure has been enlarged for better visibility, and the previously incorrect time axis labeling has been corrected.

[Figure]

**Figure 9: Comparison of synthetic and field seismic data. (a) Shot gather at the same location for synthetic (left) and 21YY field data (right). (b) PSTM comparison between the synthetic model and the 21YY line. (c) Enlarged views of discontinuous reflections (synthetic, field). (d) Comparison of dipping bed reflections (synthetic, field), showing shadow zones and steep basal topography.**

10. On figure 9, again, the same issue with reflection coefficient used as the colorbar. Neither the migrated sections nor synthetics are showing reflection coefficient, but rather normalized amplitude.

➔ We have revised the colorbar label from "Reflection coefficient" to "Normalized stacked energy".

11. I think this gets into a slight issue I have with the results, which is that apart from travel time calculations, they are mainly qualitative. The velocity of the subsurface is assumed (water, bed) rather than estimated via something like AVO to give you actual reflectivity. This means that while yes, the synthetics have the same shape, their amplitudes are not consistent with each other both laterally and in time (later arrivals are significantly weaker). This is likely due to a combination of attenuation and structure, but it was unclear whether this is considered in the modeling. I do feel that this comparison is only relevant for the geometrical features such as the scour surfaces or the steep bed slopes.

➔ As described in our response to Comment 4, there are inherent physical limitations in conducting a reliable AVO analysis in this study. Due to the restricted offset range of the acquired data, we instead constructed a physically reasonable subsurface velocity model informed by previous literature and the geological context of the study area. Based on this model, we generated synthetic seismograms. We fully acknowledge that the absolute amplitudes of the synthetic data may not perfectly match those of the field data. Therefore, rather than focusing on amplitude strength, the primary objective of our interpretation is to conduct qualitative structural analysis using the wave polarity and arrival times of key reflections, including reflected (PP) waves, ghost (pPP) waves, and lake-bottom reflected (PPPP) waves, as well as reflections from erosional surfaces and steep bedrock interfaces. Quantitative analysis is limited to depth estimation based on direct and refracted wave velocities.

The observed weakening of the ghost signal amplitudes is attributed to high attenuation within the firn layer near the surface, which cannot be explicitly modeled in our velocity structure due to its complex and variable physical properties. Consequently, some mismatch in amplitude is expected between the field and synthetic data.

Nevertheless, given the acquisition conditions, the qualitative match between synthetic and observed waveforms is enough to support the geometric interpretations in this study — specifically about ice thickness, lake depth, scour morphology, and steep basal boundaries. Therefore, our quantitative conclusions are limited to those parameters derived from geometry.

12. Additionally, why are the timings of the synthetics and field data off? The synthetics arrive consistently earlier than the observations. Does this mean the depth used is too shallow? Or the firn is not fully accounted for?

➔ The observed discrepancies arise in part from our use of a homogeneous synthetic model based on a representative velocity structure estimated from Shot Point 1, applied uniformly across the entire 21YY survey line. While the travel times at Shot Point 1 closely match the observed data, deviations may occur at other locations along the line, where travel times may appear faster or slightly misaligned. These discrepancies are primarily attributed to uncertainties in the thickness and velocity of the firn and ice layers. As noted earlier, we are currently conducting additional data analysis and high-resolution velocity inversion to enable the application of an improved firn velocity model. This refinement will be incorporated in future work to enhance the accuracy of travel time predictions and waveform matching.

13. Generally, I am not exactly sure how the results compare to the previous estimates of the D2 boundaries. From my understanding, the seismic interpretation is that the lake is on the opposite side of the marked boundary in orange on Figure 3. This could be me confusing the direction of the cross sections, but I think that emphasis the need for more clarity about the geometry of the features described.

➔ Please refer to our response to the comment 2, where this issue is addressed in detail.

14. Figure 10: Could you provide a reference for the resolution of both the IPR and seismic data? Perhaps on the plot, add points for the individual datapoints for the IPR line, which looks sparser? Additionally, why is the seismic prediction almost always slightly thinner, on the order of tens of meters? I wonder if including the uncertainty bars would help show the agreement between the two.

➔ We appreciate the reviewer's comment. As the IPR (Ice-Penetrating Radar) survey line does not exactly coincide with the seismic survey line, it is not possible to present point-by-point comparisons of the raw data from both datasets. The IPR data used in this study is an interpolated 2D profile, and values were extracted specifically at the locations corresponding to the seismic survey line. Therefore, the spatial sampling intervals between the two datasets are consistent. It is also important to note that radar wave velocity in ice is extremely high ($V \approx 0.17$ m/ns), which results in a larger inherent uncertainty in depth estimation from the IPR data. To account for this, we have included the estimated uncertainty ranges for both the IPR and seismic data in the revised figures.

[Figure]

**Figure 10**

15. I think the use of published values for glacial firn, ice, water is fine for a starting model, but there is significant variation across estimates of these values (Yang et al. 2024; Picotti, Carcione, and Pavan 2024; Agnew et al. 2023). Also, these values or at least estimates of them should be visible during the velocity analysis stage. You could get a P wave velocity from the PP observation on the velocity/depth section and use that for the ice velocity in your model rather than a published one. Perhaps they will agree, but it's worth checking. Also, firn velocities for P wave can dip below 1 km/s (Yang et al. 2024), perhaps this can help explain the timing discrepancy you observe? Again, I wonder if you can get an estimate for the shallow velocity from the observed surface waves or pseudoacoustics.

➔ Please refer to our response to the comment 7, where this issue is addressed in detail.

My suggestions for revisions are as follows:

16. Data presentation and Interpretation: Provide a figure either in the main text or supplement that shows the clearest example of the reverse polarity reflection, perhaps accompanied by a synthetic. I think that part of the confusion is the difficulty in making out features on the record section as well as the slight inconsistencies in terminology. If you can also show the normal polarity reflection from the glacier-bed clearly, perhaps showing the actual waveform as done in 9a, the difference between the two regions would be immediately clear.

➔ Please refer to our response to the comment 9, where this issue is addressed in detail.

17. Modeling: Calculate Vp, Vs in firn and ice. It doesn't have to be in depth, as I understand that is not the point of the paper, but an average value or upper/lower bounds to at least confirm that your simple model is appropriate would help strengthen the interpretation and perhaps resolve some of the discrepancies between synthetics and observations. If you don't want to derive these values from the data, I suggest trying different basal models (wet sediment/till) to confirm that your simple 2d model matches the data best.

➔ Please refer to our response to Comment 7, where this issue is addressed in detail. The velocity model was constructed using P-wave (Vp) velocities derived from field data and supported by values reported in previous studies. Due to limitations in the field data, it was not feasible to reliably extract S-wave (Vs) velocities; therefore, the synthetic model in this study was generated using Vp information only.

18. Supplementary information: Providing supplementary information, particularly on the data processing and interpretation, will clear up a lot of uncertainty about the signals that are observed. For example, the scour features; while you mention removal of surface waves and surface scattering, the detail given is not sufficient to explain exactly what the resultant signal we are seeing here is. Showing the raw data, or a panel with the data for each or important processing steps could help clarify your certainty in interpreting these features as structural rather than artifacts from the processing. I believe that at minimum a table which describes the processing parameters (anomalous amplitude attenuation, curvelet filtering, surface consistent amplitude compensation) is necessary for the sake of repeatability. Supplementary information can also contain a more thorough exploration of the model space.

➔ We have added additional results to the Supplementary Materials to show the outcomes at each stage of the data processing workflow. These additions help to further demonstrate that the observed scour structure is not an artifact caused by surface waves, surface scattering, or processing-induced effects, but rather represents a genuine subsurface structural signal. In addition, we have included the key processing parameters used at each step to enhance transparency and reproducibility.

Line Specific Comments within attached PDF.

➔ As the PDF file was missing and was not uploaded despite our request, we were unable to address the line comments. We have revised the manuscript and responded to the main comments raised above. If there are any important issues in the line comments that should be addressed in the manuscript, please notify us.

**References:**

Agnew, Ronan S., Roger A. Clark, Adam D. Booth, Alex M. Brisbourne, and Andrew M. Smith. 2023. "Measuring Seismic Attenuation in Polar Firn: Method and Application to Korff Ice Rise, West Antarctica." Journal of Glaciology 69 (278): 2075–86. https://doi.org/10.1017/jog.2023.82.

Horgan, Huw J., Laurine van Haastrecht, Richard B. Alley, Sridhar Anandakrishnan, Lucas H. Beem, Knut Christianson, Atsuhiro Muto, and Matthew R. Siegfried. 2021. "Grounding Zone Subglacial Properties from Calibrated Active-Source Seismic Methods." The Cryosphere 15 (4): 1863–80. https://doi.org/10.5194/tc-15-1863-2021.

Ju, H., Kang, S., Han, H., Beem, L. H., Ng, G., Chan, K., Kim, T., Lee, J., Lee, J., Kim, Y., and Pyun, S.: Airborne and Spaceborne Mapping and Analysis of the Subglacial Lake D2 in David Glacier, Terra Nova Bay, Antarctica, J. Geophys. Res.: Earth Surf., 130, https://doi.org/10.1029/2024jf008142, 2025.

Peters, L. E., S. Anandakrishnan, C. W. Holland, H. J. Horgan, D. D. Blankenship, and D. E. Voigt. 2008. "Seismic Detection of a Subglacial Lake near the South Pole, Antarctica." Geophysical Research Letters 35 (23). https://doi.org/10.1029/2008GL035704.

Picotti, Stefano, José M. Carcione, and Mauro Pavan. 2024. "Seismic Attenuation in Antarctic Firn." The Cryosphere 18 (1): 169–86. https://doi.org/10.5194/tc-18-169-2024.

Yang, Yan, Zhongwen Zhan, Martin Karrenbach, Auden Reid-McLaughlin, Ettore Biondi, Douglas A. Wiens, and Richard C. Aster. 2024. "Characterizing South Pole Firn Structure With Fiber Optic Sensing." Geophysical Research Letters 51 (13): e2024GL109183. https://doi.org/10.1029/2024GL109183.

*Supplement of*

**Seismic data analysis for subglacial lake D2 beneath David Glacier, Antarctica**

Hyeontae Ju, Seung-Goo Kang, et al.

*Correspondence to:* Seung-Goo Kang (ksg9322@kopri.re.kr)

The copyright of individual parts of the supplement might differ from the article license.

S1. Seismic data processed parameters and results

This study utilized the Omega geophysical data processing platform (SLB) for seismic data processing. Among the various processing steps, we provide below the key parameters applied during procedures that directly influence the ice–bedrock interface signal, such as noise attenuation.

1.   Anomalous amplitude attenuation (AAA) for the 1$^{st}$ round

AAA is a frequency-domain filtering technique designed to suppress spatially coherent anomalous amplitudes such as swell noise and rig noise, by comparing amplitude spectra across traces and attenuating outliers based on spatial median statistics.

The method identifies frequency bands with anomalous energy by comparing each trace's amplitude spectrum within a spatial window to the median of its neighboring traces. Detected anomalies are either scaled or replaced using interpolated values from adjacent traces, preserving relative amplitude relationships. Key parameters include TIME, which defines the temporal window of threshold application; THRESHOLD FACTOR, which sets the amplitude level considered anomalous; and

SPATIAL MEDIAN WIDTH, which specifies the number of adjacent traces used for median computation. Proper tuning of these parameters is essential to avoid signal distortion while effectively attenuating coherent noise. AAA is particularly useful in prestack data conditioning as it enhances seismic data quality without compromising true subsurface reflections (SLB,

2025a).

●   SPATIAL MEDIAN WIDTH: 21 traces

●   Threshold factor tables:

| TIME | THRESHOLD FACTOR |
|---|---|
| 0 | 15 |
| 1000 | 10 |
| 3000 | 7 |
| 4000 | 6 |

2.   Curvelet transform-based filter for 1$^{st}$ round (Figure S43b)

Curvelet Transform is a multi-scale, multi-directional decomposition technique that provides a sparse representation of seismic data by capturing curved wavefronts more efficiently than conventional fourier or wavelet transforms. An important aspect of the Curvelet Transform implementation involves user-defined control over the scale and angle bounds that determine which components of the data will be transformed. The LOWER BOUND OF SCALE and HIGHER BOUND OF SCALE specify the range of spatial frequencies (scales) to be included in the transform. Lower scales correspond to coarse, low-frequency components, while higher scales capture fine, high-frequency structural details. The LOWER BOUND OF ANGLE and

HIGHER BOUND OF ANGLE define the directional sectors (angles) within each scale to be analyzed. This allows selective enhancement or suppression of events based on their dip or propagation direction (SLB, 2025b). Figure S1 illustrates how the f–k domain is partitioned into curvelet panels by scale and angle. Adjusting these bounds allows for targeted signal processing, such as isolating curved events or attenuating directionally coherent noise. These parameters provide valuable flexibility in customizing the transform for specific seismic applications.

● Panel manager

| LOWER BOUND OF THE SCALE | HIGHER BOUND OF THE SCALE | LOWER BOUND OF THE ANGLE | HIGHER BOUND OF THE ANGLE |
|---|---|---|---|
| 2 | 2 | 1 | 3 |
| 2 | 2 | 8 | 10 |
| 3 | 3 | 1 | 6 |
| 3 | 3 | 13 | 18 |
| 4 | 4 | 1 | 6 |
| 4 | 4 | 13 | 18 |

[Figure]

**Figure S1: Illustration of the panel manager. In the f-k domain, the hatched area is identified as noise and removed accordingly.**

3. Surface-consistent deconvolution

Surface-Consistent Deconvolution is a technique for generating and applying deconvolution operators that are consistent across seismic sources, receivers, offset ranges, and CMP locations (SLB, 2025c; Yilmaz, 2001).

Key processing parameters used in this workflow include:

● CONSTANT_ACOR_LENGTH = 100: Defines the half-length of the autocorrelation window used in operator design, balancing spectral resolution and filter stability.

●   WHITE NOISE PERCENT = 0.01: Adds 1% white noise to stabilize the autocorrelation estimation and prevent
over-whitening of the signal.
●   PREDICTION DISTANCE = 2.5: Specifies the prediction lag in the predictive filter design; this parameter
controls the temporal range of the filter's effect, influencing multiple suppression and resolution.

4.   Anomalous amplitude attenuation (AAA) for the 2nd round
●   Spatial median width: 11 traces
●   Threshold factor tables:

| Time | Threshold factor |
|------|------------------|
| 0 | 8 |
| 1000 | 6 |
| 3000 | 4 |
| 4000 | 3 |

5.   Curvelet transform-based filter for the 1st round: same as 1st round parameter

6.   Frequency-offset coherent noise suppression (Figure S3.c)
The frequency–offset (F-X) Coherent Noise Suppression (FXCNS) module is designed to attenuate near-surface shot-
generated coherent noise, such as dispersive surface waves and trapped modes, which interfere with primary seismic reflections,
particularly in 3D shot or receiver gathers with irregular spatial sampling (Hildebrand, 1982). FXCNS operates in the frequency
domain by modeling coherent noise using fan filters and estimating it in a least-squares sense for each trace based on local
neighbors within a specified azimuthal sector. The estimated coherent noise is then subtracted from the original signal,
preserving true reflection events (SLB, 2025d).
●   LOW PASS VELOCITY: 100
●   LOW STOP VELOCITY: 300
●   HIGH PASS VELOCITY: 8000
●   HIGH STOP VELOCITY: 10000

[Figure]

        (a) Raw data shot gather of synthetic              (b) After pre-stack time migration

**Figure S2: Results before and after data processing. (a) Synthetic raw data of shot gather #1. (b) Result after pre-stack time**
**migration.**

[Figure]

**Figure S3: Results at each stage of data processing. (a) Shot gather #1 from 21YY. (b) Removal of high-frequency random noise and coherent linear noise. (c) Application of a frequency-offset coherent noise filter and tau-p linear noise attenuation for surface wave removal. (d) Result after applying pre-stack time migration.**

**References**

Hildebrand, S. T.: Two representations of the fan filter, Geophysics, 47, 957–959, https://doi.org/10.1190/1.1441363, 1982.

SLB: *Omega Geophysical Processing System – Anomalous Amplitude Noise Attenuation (ANOMALOUS_AMP_ATTEN)*
*Module Documentation*, Version 27.14, SLB manual, Houston, TX, 2025a.

SLB: *Omega Geophysical Processing System – Curvelet Transform (CURVELET_TRANSFORM) Module Documentation*,
Version 22.1, SLB manual, Houston, TX, 2025b.

SLB: *Omega Geophysical Processing System – Surface-Consistent Deconvolution Analysis (SC_DCN_SPCTRL_ANL)*
*Module Documentation*, Version 13.18, SLB manual, Houston, TX, 2025c.

SLB: *Omega Geophysical Processing System – F-X Coherent Noise Suppression (FXCNS) Module Documentation*, Version
3.16, SLB manual, Houston, TX, 2025d.

Yilmaz, Ö.: Seismic data analysis: Processing, Inversion, and Interpretation of Seismic Data, Appendix B.8 (Surface-
Consistent Deconvolution), Society of Exploration Geophysicists, 262–266 pp., 2001.

---

## Referee Report (RR2)

[referee-annotated manuscript omitted]

---

## Editor Decision (ED1)

Final corrections for egusphere-2025-2055 "A seismic analysis of subglacial lake D2 (Subglacial Lake Cheongsuk) beneath David Glacier, Antarctica"

L224: "approximately 2.01 m to 5.27 m." Quoting these values to two decimal places doesn't sound very approximate! Might it be better to report them as "approximately 2.0 m to 5.3 m"?

Figure 6 – label "Discontinous" should be "Discontinuous"

L372-373, L377: is precision of two decimal places really justified for these errors? Would integers be more defensible?

---

## Author Response (AR2)

Referee #1

My main concerns centre around Figures 4 and 6.

Ice base reflection

In figure 4 the ice-base reflection is shown in panels c and f. These plots are still too zoomed out to adequately interpret. It would be better to zoom in on a 50-100 ms window that encompasses the ice-water interface and its ghost. This would allow the reader to assess the polarity, and any possibility of a mixed phase return resulting from either thin water or saturated sediments, with the later reflection indicating the base of the sediments. I don't expect the reflections to resemble ricker wavelets but significant mixed phase may suggest thin layer effects. I don't suggest this is the case but as the results presented here may guide a drilling program it pays to proceed with an abundance of caution.

More importantly, the theoretical ice-water amplitude versus offset curve has peak (-ve) amplitude at zero offset, decreasing to zero at ~60 degrees then increasing again. That's not what these reflections appear to do. This may be a result of a processing step, or may indicate the presence of something other than water at the bed. Again, I don't suggest this is the case, but it would be good to present the data in a way that allows this to be assessed. This difference is again highlighted in the synthetic—field comparison in Figure 9a where the field data show low amplitudes at near offsets.

➔ Before proceeding with our response, we have redefined the symbols to avoid confusion. We have added Table 2 to the manuscript.

**Table 2: Symbols for each reflection event**

| Interface symbols | Model 1 | Model 2 |
|---|---|---|
| ① | Ice-water | Ice-water |
| ② | Ice-water ghost | Ice-water ghost |
| ③ | - | Water-sediment |
| ④ | - | Water-sediment ghost |
| ⑤ | Water-bed | Sediment-bed |
| ⑥ | Water-bed ghost | Sediment-bed ghost |
| ⑦ | Ice-bed | Ice-sediment |
| ⑧ | Ice-bed ghost | Ice-sediment ghost |

[Figure]

[Figure]

**Figure R 1. Velocity model for the subglacial lake structure interpretation. Boundary numbers correspond to those indicated in the manuscript, and the theoretical reflection coefficients at each interface are shown.**

➜ We identified an error in the display of near-offset signal weakening in the field data during the Python-based rendering process. We appreciate the reviewer's observation, which helped us identify this issue. Accordingly, Figures 4c, 4f, and 9a have been updated. In response to the reviewer's insightful suggestion, we agree that the presence of subglacial sediments beneath the lake provides a valuable interpretative framework. To explore this possibility, we propose an additional model (Model 2) that includes a sedimentary layer beneath the subglacial lake, in addition to the original ice–water–bedrock model (Figure R1). We extended our interpretation to include a weak reflection approximately 11 ms below the ice–water interface, which may represent the water–sediment boundary. However, in some areas, signals attributable to the sediment interface are not clearly observed. Considering this interpretational ambiguity regarding the sediments, we compared two structural models with the field data. Accordingly, we have revised Section 5 of the revised manuscript and Figure 8 as follows.

(Page 14, Lines 277-289) "In all seismic profiles, the glacier–water interface (①) is characterized by strong, reverse polarity reflections. Following this, a relatively weaker reflection (③) with limited lateral continuity, which may indicate an unconsolidated sediment layer, or an unknown interface beyond the scope of current interpretation. […] To address these challenges, this study developed a subsurface structural model and conducted a comparative analysis of synthetic seismograms generated from the model with observed field data. Focusing on the interpretation of basal reflections beneath the subglacial lake—excluding the glacier–lake interface (①)— two plausible structural models were proposed. Model 1 assumes the absence of a sedimentary layer, in which reflection (③) is not present, and reflection (⑤) represents the base of the subglacial lake. In contrast, Model 2 includes a sedimentary layer, where reflection (③) corresponds to the lake–sediment interface and reflection (⑤) indicates the sediment–bedrock interface (Figure 8)."

[Figure]

**Figure 8: P-wave velocity model used in forward modeling for line 21YY. The upper ~100 m represents firn with velocities ranging from 1525–3800 m s⁻¹ (Kirchner and Bentley, 1979; Picotti et al., 2015; Qin et al., 2024). The ice below this depth has a velocity of 3800 ± 5 m s⁻¹ (Kohnen, 1974), and the subglacial water layer has a velocity of 1396 ± 2 m s⁻¹ (Thoma et al., 2010; Tulaczyk et al., 2014). In Model 2, the velocity of 2817 m s⁻¹ for the sediment layer was taken from the lower sediment layer model of Lake Vostock (Carcione & Gei, 2003).**

➜ In addition, regarding AVO analysis—a method for quantitative interpretation—we note that the maximum usable angle in this dataset is limited to approximately 36° due to acquisition geometry (Figure R2), which prevents a clear observation of amplitude variation with offset (-ve -> 0 -> +ve). We computed theoretical AVO curves using estimated material properties for each layer, and overlaid amplitude values extracted by picking the ice–water interface in Shot Gather #2 of the 21YY line. The observed AVO trend for the ice–water interface is consistent with theoretical predictions. However, since this comparison is based on a single-shot gather, a comprehensive analysis would require significant additional processing. We are currently conducting further research—such as velocity optimization in the firn layer, deghosting, and extended AVO analysis—with the intention of presenting these findings in a future publication.

[Figure]

**Figure R 2. Reflection amplitude versus incidence angle curves for the ice–water boundary (①) from Shot gather #2 of the 21YY line and for each media interface.**

Sub ice-base reflection

Figure 6 raises the possibility that the currently interpreted water-bottom (floor of the subglacial lake) reflector may be deeper than the actual lake bottom. The grounds for this come from the presence of reflectivity between the ice-base reflector and its ghost.

For example, in Figure 6a) the reflectivity between the ice-base (1) and the ghost (2) needs explaining. If (1) is an ice-water interface is this reflectivity the base of the water column? Why is (3) the preferred water column base? The reflectivity at distance 2.5-3 km that impinges on the basal return looks to me like an ice-bed reflection. Again in Figure 6b, the reflectivity between the bed return and the ghost needs to be explained. Is it possible that this is the lake bottom instead of (3)? In Figure 6c between ~2.7 km and ~4 km there is a prominent reflector arriving before the ghost. How can this reflector be explained? Why is the deeper reflector the preferred lake bottom? As the survey will likely guide site selection for a subglacial access program it would pay to clearly outline the reasoning behind the preferred interpretation, and provide alternative interpretations.

➜ We present both the original and additional interpretations in Figure R1. The weak reflection observed between the ice–water interface and its corresponding ghost may represent the water–sediment boundary. Given that the uppermost part of the sediment layer is likely unconsolidated, resulting in low reflection amplitudes, this interpretation is considered more plausible. Accordingly, we have introduced Model 2 in the manuscript and conducted a comparative analysis using synthetic data. We have revised Section 4 of the revised manuscript as follows.

(Page 11, Lines 231-233) "Between reflections ① and ②, a weak normal polarity reflection (③), presumed to represent an interface, is observed. However, in some shot gathers, signal ③ appears with reverse polarity (Figure 4c), leading to partial cancellation and ambiguity in layer interpretation. Approximately 25 ms later, an opposite polarity ghost reflection (④) follows."

Minor/technical points.

Title. I suggest a change to "A seismic analysis of subglacial lake D2 (Subglacial Lake Cheongsuk) beneath David Glacier, Antarctica."

➜ Thank you. We have changed the title.

L34-35 'largely isolated' Not an important point, but I still don't think this can be concluded. A stable lake just implies steady-state where inputs and outputs are balanced.

➔ (Page 2, Lines 36-38) We have revised the sentence from

"These closed systems do not exhibit significant surface elevation changes and where subglacial water remains largely isolated, with minimal exchange due to slow and stable recharge and discharge cycles" to

"These closed systems do not exhibit significant surface elevation changes and are characterized by long-term balance between recharge and discharge, although the extent of subglacial water exchange remains uncertain in the absence of direct observations".

L106 '...presence of subglacial sediments' I think a reference is needed here. Perhaps https://doi.org/10.1130/G50995.1

➔ (Page 5, Line 113) We have added the reference "(Siegfried et al., 2023)"

L113-115. Please break up this sentence. The meaning in currently unclear.

➔ (Page 5, Lines 120-122) We have revised the sentence from

"To better constrain the lake's extent and basal conditions of SLD2, airborne IPR survey data from 2016/17 (Lindzey et al., 2020) and 2018/19 (Ju et al., 2025) field campaigns indicate that glacier surface elevations in the SLD2 region range from approximately 1820 to 1940 m, with ice thicknesses varying between 1685 and 2293 m" to

"To better constrain the extent and basal conditions of SLD2, we used airborne IPR data collected during the 2016/17 (Lindzey et al., 2020) and 2018/19 (Ju et al., 2025) field campaigns. These surveys show that the glacier surface elevation in the SLD2 region ranges from approximately 1820 to 1940 m. The corresponding ice thicknesses vary between 1685 and 2293 m".

L117 'Bain-like topography' is not a well-known term in glaciology. Please define.

➔ (Page 5, Line 124) Sorry, it's a typo. We have revised the from "Bain-like" to "basin-like".

L127 'were aligned' sounds deliberate, I suggest 'happened to be aligned'.

➔ (Page 5, Line 134) Thank you. We are not deliberate. We have revised from "were aligned" to "happened to be aligned".

L132 'reduces' -> 'reduced'

➔ (Page 6, Line 138) We have revised from "reduces" to "reduced".

Figure 3. The seismic lines in this figure need distance annotations so that the seismic stacks can be referenced to the basemap.

➔ We have added the length at the end of each seismic line.

[Figure]

**Figure 3**

Table 1 (or the text) requires additional details of shot positioning (off-end, centre shots?) and near offset distance.

➔ (Table 1) We have added the details of shot positioning.

(Page 8, Line 163) We have added the sentence "Detailed shot positioning information is provided in supplementary information S1".

**Table 1: Parameters of the active-source seismic survey.**

| Survey Parameters | Survey lines | | | |
|---|---|---|---|---|
| | 21X line | 21Y line | 21XX line | 21YY line |
| Line length (km) | 5 | 3.5 | 5 | 3.5 |
| Fold | 8 | 8 | 4 | 4 |
| Shot interval (m) | 90 | 90 | 180 | 180 |
| Number of shots | 56 | 40 | 28 | 20 |
| **Shot positioning** | **Use both off-end and center shots** | | | |
| Receiver channels | 96 | | | |
| Receiver interval (m) | 15 | | | |
| **Near offset (m)** | **0** | | | |
| **Far offset (m)** | **1425** | | | |
| Recording time (s) | 4 | | | |
| Record peak frequency (kHz) | 1 | | | |
| Record sampling rate (ms) | 0.25 | | | |

| Survey time (days) | 34 |
|---|---|
| Survey crew size | Hot water drilling (3), Seismic (6) |

➔ Supplementary information S1. Seimsic data acquistion

[Figure]

**Figure S1: In the seismic survey layout, only the odd-numbered receivers are displayed, that is, one receiver marked every two channels.**

L160 Citing the Voigt publication on georods seems more appropriate here.

➔ (Page 8, Line 167) We have revised from "(Ju et al., 2024)" to "(Voigt et al., 2013)".

Figure 4. b) I appreciate the lines are indicative only and not supposed to represent picks, but the shallow gradient on the direct arrival implies a very high velocity. I suggest changing the gradient to one more representative of the velocity estimated.

➔ The slope lines have been adjusted to reflect the correct velocities.

L211-212 'This resolution is adequate...' I don't think this statement is needed. The data are what they are and are capable of imaging the top and bottom of an approximately 2 m thick water column.

➔ (Page 11, Line 224) We have revised the sentence from "This resolution is adequate for imaging SLD2" to "The data can image both the top and bottom of a water column approximately 2 m thick or thicker.".

Figure 6 caption: Include comment in caption that the annotations are discussed in the main text.

➔ (Figure 6 caption) We have added the sentence "See Table 2 for symbols definitions".

L352—354 I think a more nuanced description is required here.

➔ (Page 20, Lines 397-400) We have revised the sentence from

"The seismic data revealed strong, laterally continuous reflections with reverse polarity at the glacier–lake

interface, whereas normal-polarity reflections were observed at the glacier–bed and lake–bed interfaces." to

"The field seismic data revealed strong, reverse polarity reflection at the glacier–lake interface. In contrast, the basal reflections beneath the lake are less well-defined, suggesting the presence of subglacial sediments. This ambiguity gives rise to two alternative interpretive scenarios based on the presence or absence of a sedimentary layer.".

L358 Again, I worry that the drilled water depths could be much thinner than this if the reflection events that are evident between the primary bed return and the ghost or the bed returns are in fact the base of the water column.

➔ (Pages 20-21, Lines 401-417) We have revised the sentence from

"A comparison between synthetic and field PSTM sections demonstrated strong agreement in the timing and polarity of major reflection events at the glacier-lake and lake-bed interfaces, confirming the validity of the velocity model. This model estimated the ice thickness and lake water column height to be 2250–2300 m and 53–82 m, respectively." to

"Given this interpretational ambiguity regarding the sediment layer, two velocity models were constructed: Model 1, which assumes the absence of sediment, and Model 2, which includes a sediment layer beneath the lake. Synthetic seismology was generated using wave propagation modeling based on these models. Sediment thickness in Model 2 was uniformly assigned using the average time difference ($①$–$③$) calculated from selected areas of the dataset. Comparisons between the synthetic and field PSTM sections show consistent TWT times and polarities for key reflection events at the glacier–lake interface ($①$), the lake–bedrock interface ($⑤$) in Model 1, and the sediment–bedrock ($⑤$) interface in Model 2. […] Furthermore, the integrated analysis of seismic and synthetic data provides a quantitative structural model of the SLD2-A geometry beneath David Glacier. These results provide critical guidance for future clean hot-water drilling. In particular, we identify an area within a 1 km radius of S 75.422°, W 155.441° as a suitable candidate site, based on its broad spatial extent, minimum estimated water depth exceeding approximately 10 m, and absence of contamination from surface field camps.".

L367. Regarding the suggested drill site, this should be included on a previous basemap and the corresponding seismic profile and distance marker referenced here.

➔ (Figure 3) We have revised the map in Figure 3, and the candidate drilling site is marked.

Referee #2

1. Figure 6d: I see how the reflections 3/4 on the left side of the image 6d show normal then reverse polarity respectively, but on the right side, the polarity is flipped for 3 and 4. The reflection claimed to be from the ice-bed interface at 3 km on line 21 YY is negative polarity then positive polarity, which is opposite to what is described in the text and what would be expected for ice-rock. I mentioned this in the first submission but it may not have been seen by the authors. Clearly, given the geometry of the reflection, arrivals 3 and 4 have to represent the edge of the subglacial lake. However, following the logic of the authors, from the polarity I would assume this region too has subglacial water, or at least material with lover velocity than the ice above. If the authors are certain this region should have bedrock or lithified sediment, then it brings question to the use of polarity on its own to describe the subsurface velocities and materials.

➔ Before proceeding with our response, we have redefined the symbols to avoid confusion. We have added Table 2 to the manuscript.

**Table 2: Symbols for each reflection event**

| Interface symbols | Model 1 | Model 2 |
|---|---|---|
| ① | Ice-water | Ice-water |
| ② | Ice-water ghost | Ice-water ghost |
| ③ | - | Water-sediment |
| ④ | - | Water-sediment ghost |
| ⑤ | Water-bed | Sediment-bed |
| ⑥ | Water-bed ghost | Sediment-bed ghost |
| ⑦ | Ice-bed | Ice-sediment |
| ⑧ | Ice-bed ghost | Ice-sediment ghost |

[Figure]

**Figure R1. Velocity model for the subglacial lake structure interpretation. Boundary numbers correspond to those indicated in the manuscript, and the theoretical reflection coefficients at each interface are shown.**

At the 3 km of the 21YY line, reflections ③ (redefined as ⑦) and ④ (redefined as ⑧) exhibit reverse and normal polarity, respectively. In line with the reviewer's suggestion, we acknowledge that this interface may represent an ice–sediment boundary rather than an ice–bedrock boundary. Booth et al. (2012) showed that when a dilatant till exists as a thin layer, it can give rise to reverse polarity reflections. The observed phase reversal, therefore, strongly suggests the presence of a dilatant till at the ice–sediment interface. By contrast, our synthetic model, which does not include a till layer, produces normal polarity reflections. We simplified the synthetic modeling and incorporated sediment properties predicted for Lake Vostok. However, because the physical parameters of materials such as dilatant till span a wide range and site-specific values are difficult to constrain, it is challenging to include them directly in the modeling. Accordingly, to clarify that the observed reverse polarity can be plausibly explained by the presence of subglacial till, we have added the following sentence in section 5 of our revised manuscript.

(Pages 18-19, Lines 361-366) "However, the field data show that reflection ⑦ exhibits reversed polarity,

suggesting the presence of subglacial sediments with lower acoustic impedance than assumed in the models. This discrepancy may be explained by the presence of a dilatant till beneath the glacier, which can produce reverse polarity reflections depending on its physical properties. Booth et al. (2012) demonstrated that the seismic response of such tills is highly sensitive to variations in P-wave velocity, density, and thickness. In particular, their study showed that when the till forms a thin layer, reverse polarity reflections may occur."

(Pages 20-21, Lines 406-411) "Nevertheless, synthetic data generated by modeling a velocity model that simplifies a complex geological structure has limitations in thoroughly explaining the entire waveform of the complex field data. For example, subglacial sediments are generally expected to produce normal polarity reflections due to acoustic impedance contrasts with overlying water. However, in field data, the polarity and clarity of the water-sediment interface vary with the degree of sediment consolidation. In particular, the reverse polarity reflection observed at the ice–sediment interface in the 21YY profile suggests the potential presence of dilatant till."

2. Figure 9: This issue is even more prominent here, and hasn't changed since the first submission. In figures 9c and 9d, arrivals 3 and 4 in the synthetic data appear to be the opposite polarity than what is observed in the field data. The strongest first reflection in 9d corresponding to the field data appears to be blue with red side lobes (negative polarity) while the synthetic shows red with blue side lobes (positive polarity). This is true for 9c as well, where the lake-bed interface is supposed to show normal polarity corresponding arrival 3 based on the synthetics, but in the field data this arrival is negative polarity. Further, it appears that there is a transition where at 0.4 km, the arrival 3 is normal polarity, but by 1.2 km, the polarity has flipped. Whether this is a geometrical eTect (unlikely since oTsets are small) or a difference in material (hard rock vs wet sediment), this needs to be addressed. Alternatively, the argument can be strengthened with evidence besides polarity of the first arrival. If you are able to perhaps compute reflectivity curves or compare it to later phases' polarity (PS, SS etc.), these discrepancies between the data and synthetics may be reconciled. As it stands, the simple velocity model and generated synthetic wavefield cannot describe the full wavefield of the field data.

➔ The explanation regarding the reverse polarity of reflections ⑦ (pre-revision ③) and ⑧ (pre-revision ④) has been addressed in Response 1.
For reflection ⑤ (pre-revision ③) observed in the 21YY section, we maintain that it is a normal polarity event consistent. In the final migrated image—particularly after residual static correction—we acknowledge that the red amplitude (the black arrow) above the side lobe (blue) appears prominent and that the peak (red) can appear less prominent than the side lobe (blue) (Figure R2). This visual ambiguity may, at first glance, make the reflection look like a reverse-polarity event. To aid interpretation and resolve this ambiguity, we have added a guideline (black dashed line) in Figure 6d that explicitly indicates our interpretation of the normal polarity peak of reflection ⑤.

[Figure]

**Figure R2. Comparison of the lower reflection signal before and after residual static correction.**

[Figure]

**Figure 6: PSTM seismic sections for lines (a) 21X, (b) 21Y, (c) 21XX, and (d) 21YY prior to ghost removal. Ghost reflections appear 25–30 ms beneath the glacier–lake and lake–bed interfaces due to the 25 m source depth. See Table 2 for symbols definitions.**

3. As an overall note, the field data is clearly more complicated than the synthetic data. For the broad identification of arrivals and discontinuous features, the synthetic data seems to do a good job in guiding one based on qualitative similarity to the field data. However, the actual arrivals, both in terms of polarity but also in terms of coherence, can be quite diTerent. Use 9d as an example. In addition to being diTerent polarity than the synthetic, why does the reflection become discontinuous laterally? Is it data quality, subsurface topography, variable materials properties? While the "scour" related discontinuities are explained by c and d on labeled on figure 9, there is significant incoherence and amplitude variation laterally in addition to these discontinuities.

➔ Lateral reflection discontinuities observed beneath the glacier may result from subsurface topographic variations—such as subglacial features like the SLF—as well as from data quality issues or spatial heterogeneity in material properties. In this context, topographic variations beneath the surface are functionally equivalent to lateral variations in physical properties. In this study, we carefully assessed the quality of all raw data and completed the preprocessing and validation procedures. Elevation corrections were also performed to minimize errors associated with surface topography. Events ⓐ, ⓒ, and ⓓ appear to be caused by the loss of coherent reflections due to structural scattering along inclined surfaces. Accordingly, the lateral discontinuities are interpreted as expressions of subglacial terrain changes, such as the SLF, which reflect material contrasts. In the case of ⓑ, the feature resembles a bow-tie pattern and may indicate a protruding ice structure beneath the glacier, as in Figure 7.

4. In terms of resolution, the authors discuss the vertical resolution of the water layer. However, in terms of lateral resolution, the discussion is a bit lacking. If, for example, there are small pockets of water on the ice-bed region at higher elevations, how would this affect the polarity and coherence of the arrivals here? Would they be resolved at all? Similarly, at the ice-lake-bed interface, can discrepancies in synthetics and observations be explained by varying properties laterally, i.e. some kind of water-wet sediment distribution at the glacier base rather than purely

water?

➔ The ice–bed interface is interpreted as a gently dipping structure. If small water pockets are present, they would fall below the vertical resolution limit and thus be difficult to distinguish on the seismic section. Moreover, Ju et al. (2025) reported a low probability of subglacial water in this area, and Booth et al. (2012) showed that thin layers of till can produce reversed polarity. Accordingly, the presence of a dilatant till is the more plausible interpretation. The lateral resolution, defined by the CDP spacing, is approximately 7.5 m. Although this is finer than the vertical resolution, it is sufficient for interpreting basal topography, as demonstrated by previous seismic studies at Subglacial Lake Whillans and Lake Ellsworth. We also compared the field data with synthetic seismograms to reduce interpretational uncertainty. Admittedly, synthetic data generated from a velocity model that simplifies complex geology cannot perfectly reproduce the full waveforms of field records. Nevertheless, the principal reflection events are consistent, and the data possess adequate resolution for our interpretation.
Clean hot-water drilling is planned at the SLD2 site during the 2028/29 austral summer season. If the drilling is successful, we will install a distributed acoustic sensing (DAS) cable in the borehole to conduct a seismic tomography survey. This approach is expected to yield a more accurate structural characterization of SLD2.

Overall, I find the paper to be a great read which presents interesting and novel results from an important (and understudied) region. I think there are still some considerations missing from the interpretation and discussion sections, particularly in terms of polarity and synthetic modeling, but I believe these can be addressed with relatively minor corrections. Line specific comments and more in depth description of points above are attached in the pdf.

Minor comments

Add a note about how you deal with the crevasse noise? This couldn't go away with increased fold coverage.

➔ (Page 3, Line 78) We have added the sentence "Furthermore, the sound source was positioned further from the crevasse (end-shot), delaying the arrival of crevasse-generated noise and preventing it from obscuring key reflections".

I think this should be rephrased. Without reason to think this stability should be threatened, this seems unnecessary. Also, clarify the difference between a stable glacier but an active subglacial system, which is what you claim to have here

➔ (Page 4, Lines 97-99) We have revised the sentence from
"Although the overall mass balance of David Glacier currently appears stable, it remains uncertain how long this stability can be maintained." to
"Although the overall mass balance of David Glacier currently appears stable, several active subglacial lakes observed by satellites have the potential to influence glacier dynamics (Ju et al., 2025; Kim et al., 2025).".

I think this section can be compressed

➔ (Page 6, Lines 135-141) We have revised the sentence from
"Consequently, the acquired seismic data were significantly contaminated by strong linear coherent noise associated with crevasses, which severely degraded the signal quality of key reflectors, particularly reflections from the subglacial lake–bedrock interface. In addition, explosives are deployed within shallow boreholes (< 20 m depth), and owing to the absence of proper backfilling and the rapid timing of detonation, poor coupling between the explosives and the borehole walls further reduced energy transmission efficiency, resulting in overall low-quality reflection signals (Ju et al., 2024). As a result, due to the limitations of single-fold acquisition, stacking was not feasible, resulting in a low signal-to-noise ratio (SNR) and the presence of dominant coherent noise, rendering the seismic dataset unsuitable for quantitative structural interpretation." to
"Consequently, the acquired seismic data were significantly degraded by strong linear coherent noise generated by crevasses, severely compromising the quality of key reflectors, particularly those at the subglacial lake–

bedrock interface. Furthermore, explosives were deployed in shallow boreholes (< 20 m depth), and due to the absence of proper backfilling, poor coupling between the explosives and the borehole walls further reduced energy transmission efficiency, resulting in overall low-quality reflection signals (Ju et al., 2024). Combined with the limitations of single-fold acquisition, stacking was not feasible, the dataset exhibited a low signal-to-noise ratio (SNR) and was unsuitable for quantitative structural interpretation."

Many seismic surveys claim to need 10 + fold coverage to appropriately image/resolve subglacial features. Based on the data, it's clear 8 folds is sufficient for your analysis. Do you have a thought on this discrepancy? Are 10+ folds more than necessary?

➔ While 10+ fold coverage is generally preferred for optimal seismic imaging, such acquisition geometries are often challenging to achieve in polar environments due to logistical, meteorological, and budgetary constraints. In this study, we acquired seismic data with 4- and 8-fold coverage. Despite the relatively low fold, the data quality was sufficient to resolve key subsurface features, including those indicative of subglacial lake structures. Our interpretation is supported by analogous cases, such as Horgan et al. (2012), where subglacial lake features were successfully identified using low-fold seismic data.

Here, specifiy the refraction you believe to be observing. Refraction from the firn-ice transition?

➔ (Page 8, Line 169) We have revised the sentence "In these shot gathers, the velocity of the direct wave is estimated to be approximately 1800 m/s, and the refracted wave velocity in **firn-ice transition** is approximately 3800 m/s".

Is there a reason the linear coherent noise couldn't be removed from the previous survey? if the crevasse noise is all linear as you claim, shouldn't this be possible too?

➔ During the initial seismic survey, linear noise from surface crevasses overlapped with key lake reflections. Although attempts were made to attenuate this noise, much of the amplitude information was also removed in the process, rendering the lake reflections nearly indiscernible. Moreover, approximately half of the acquired data were severely affected by crevasse-induced noise. To mitigate this issue in the subsequent survey, the survey line orientation was reversed, ensuring that crevasse-generated noise would not coincide temporally with primary reflection events. As a result, the crevasse noise arrived at later times in most of the data, thereby preserving the quality of the primary reflections except in a limited portion of the dataset.

(Page 8, Lines 176-179) We have added the sentence
"The survey was designed to place the seismic source at a distance from crevasses, ensuring that crevasse-related noise would be recorded after the main reflections (1.1–1.3 s), thereby minimizing its impact (Figure 4a). While most data exhibit crevasse noise occurring after the main reflections, a reduction in the source–crevasse distance causes this noise to increasingly overlap with the primary arrivals, thereby complicating interpretation.".

what frequency band is the final migrated image in?

➔ The migrated data have a center frequency of approximately 180 Hz. Below is the frequency analysis result of PSTM for the 21YY line.

[Figure]

**Figure R 3. Average frequency of the entire trace (blue line).**

I see how the reflections 3/4 on the left side of the image 6d show normal then reverse polarity respectively, but on the right side, the polarity is flipped for 3 and 4. The reflection claimed to be from the ice-bed interface at 3 km on line 21 YY is negative polarity then positive polarity, which is opposite to what is described in the text and what would be expected for ice-rock.

➔ This issue has been addressed in Response 1.

Booth, A. D., Clark, R. A., Kulessa, B., Murray, T., Carter, J., Doyle, S., and Hubbard, A.: Thin-layer effects in glaciological seismic amplitude-versus-angle (AVA) analysis: implications for characterising a subglacial till unit, Russell Glacier, West Greenland, Cryosphere, 6, 909–922, https://doi.org/10.5194/tc-6-909-2012, 2012.

Ju, H., Kang, S., Han, H., Beem, L. H., Ng, G., Chan, K., Kim, T., Lee, J., Lee, J., Kim, Y., and Pyun, S.: Airborne and Spaceborne Mapping and Analysis of the Subglacial Lake D2 in David Glacier, Terra Nova Bay, Antarctica, J. Geophys. Res.: Earth Surf., 130, https://doi.org/10.1029/2024jf008142, 2025.

*Supplement of*

**A Seismic analysis of subglacial lake D2 (Subglacial Lake Cheongsuk) beneath David Glacier, Antarctica**

Hyeontae Ju, et al.

*Correspondence to:* Seung-Goo Kang (ksg9322@kopri.re.kr)

The copyright of individual parts of the supplement might differ from the article license.

9    S1. Seismic data acquisition

11    **Table S1: Seismic survey work schedules**

| Date (dd/mm/yy) | Job | Work day (day) |
|---|---|---|
| 01/12/21 | GPR survey | 1 |
| 04/12/21 – 05/12/21 | Marked shot position | 2 |
| 06/12/21 – 28/12/21 | Hot water drilling (25 m) and explosive installation (4 lines, total 144 points) | 14 |
| 12/12/21 – 02/01/22 | Seismic survey (4 lines) | 11 |

15    **Figure S1: In the seismic survey layout, only the odd-numbered receivers are displayed, that is, one receiver marked every two**

16    **channels.**

18 S2. Seismic data processed parameters and results

19 This study utilized the Omega geophysical data processing platform (SLB) for seismic data processing. Among the various

20 processing steps, we provide below the key parameters applied during procedures that directly influence the ice–bedrock

21 interface signal, such as noise attenuation.

22

23   1. Anomalous amplitude attenuation (AAA) for the 1st round

24 AAA is a frequency-domain filtering technique designed to suppress spatially coherent anomalous amplitudes such as swell

25 noise and rig noise, by comparing amplitude spectra across traces and attenuating outliers based on spatial median statistics.

26 The method identifies frequency bands with anomalous energy by comparing each trace's amplitude spectrum within a spatial

27 window to the median of its neighboring traces. Detected anomalies are either scaled or replaced using interpolated values

28 from adjacent traces, preserving relative amplitude relationships. Key parameters include TIME, which defines the temporal

29 window of threshold application; THRESHOLD FACTOR, which sets the amplitude level considered anomalous; and

30 SPATIAL MEDIAN WIDTH, which specifies the number of adjacent traces used for median computation. Proper tuning of

31 these parameters is essential to avoid signal distortion while effectively attenuating coherent noise. AAA is particularly useful

32 in prestack data conditioning as it enhances seismic data quality without compromising true subsurface reflections (SLB,

33 2025a).

34   ●  SPATIAL MEDIAN WIDTH: 21 traces

35   ●  Threshold factor tables:

| TIME | THRESHOLD FACTOR |
|------|------------------|
| 0 | 15 |
| 1000 | 10 |
| 3000 | 7 |
| 4000 | 6 |

36

37   2. Curvelet transform-based filter for 1st round

38 Curvelet Transform is a multi-scale, multi-directional decomposition technique that provides a sparse representation of seismic

39 data by capturing curved wavefronts more efficiently than conventional fourier or wavelet transforms. An important aspect of

40 the Curvelet Transform implementation involves user-defined control over the scale and angle bounds that determine which

41 components of the data will be transformed. The LOWER BOUND OF SCALE and HIGHER BOUND OF SCALE specify

42 the range of spatial frequencies (scales) to be included in the transform. Lower scales correspond to coarse, low-frequency

43 components, while higher scales capture fine, high-frequency structural details. The LOWER BOUND OF ANGLE and

44  HIGHER BOUND OF ANGLE define the directional sectors (angles) within each scale to be analyzed. This allows selective

45  enhancement or suppression of events based on their dip or propagation direction (SLB, 2025b). Figure S2 illustrates how the

46  f–k domain is partitioned into curvelet panels by scale and angle. Adjusting these bounds allows for targeted signal processing,

47  such as isolating curved events or attenuating directionally coherent noise. These parameters provide valuable flexibility in

48  customizing the transform for specific seismic applications.

49  ● Panel manager

| LOWER BOUND OF THE SCALE | HIGHER BOUND OF THE SCALE | LOWER BOUND OF THE ANGLE | HIGHER BOUND OF THE ANGLE |
|---|---|---|---|
| 2 | 2 | 1 | 3 |
| 2 | 2 | 8 | 10 |
| 3 | 3 | 1 | 6 |
| 3 | 3 | 13 | 18 |
| 4 | 4 | 1 | 6 |
| 4 | 4 | 13 | 18 |

[Figure]

50

51  **Figure S2: Illustration of the panel manager. In the f-k domain, the hatched area is identified as noise and removed accordingly.**

52

53  3. Surface-consistent deconvolution

54  Surface-Consistent Deconvolution is a technique for generating and applying deconvolution operators that are consistent across

55  seismic sources, receivers, offset ranges, and CMP locations (SLB, 2025c; Yilmaz, 2001).

56  Key processing parameters used in this workflow include:

57  ● CONSTANT_ACOR_LENGTH = 100: Defines the half-length of the autocorrelation window used in operator

58  design, balancing spectral resolution and filter stability.

- WHITE NOISE PERCENT = 0.01: Adds 1% white noise to stabilize the autocorrelation estimation and prevent over-whitening of the signal.
- PREDICTION DISTANCE = 2.5: Specifies the prediction lag in the predictive filter design; this parameter controls the temporal range of the filter's effect, influencing multiple suppression and resolution.

4. Anomalous amplitude attenuation (AAA) for the 2nd round
   - Spatial median width: 11 traces
   - Threshold factor tables:

| Time | Threshold factor |
|------|------------------|
| 0 | 8 |
| 1000 | 6 |
| 3000 | 4 |
| 4000 | 3 |

5. Curvelet transform-based filter for the 1st round: same as 1st round parameter

6. Frequency-offset coherent noise suppression (Figure S4.c)

The frequency–offset (F-X) Coherent Noise Suppression (FXCNS) module is designed to attenuate near-surface shot-generated coherent noise, such as dispersive surface waves and trapped modes, which interfere with primary seismic reflections, particularly in 3D shot or receiver gathers with irregular spatial sampling (Hildebrand, 1982). FXCNS operates in the frequency domain by modeling coherent noise using fan filters and estimating it in a least-squares sense for each trace based on local neighbors within a specified azimuthal sector. The estimated coherent noise is then subtracted from the original signal, preserving true reflection events (SLB, 2025d).
- LOW PASS VELOCITY: 100
- LOW STOP VELOCITY: 300
- HIGH PASS VELOCITY: 8000
- HIGH STOP VELOCITY: 10000

[Figure]

(a) Raw data shot gather of synthetic          (b) After pre-stack time migration

Figure S3: Results before and after data processing. (a) Synthetic data of shot gather #1. (b) Result after pre-stack time migration. Symbols (see Table 2).

[Figure]

**Figure S4: Results at each stage of data processing. (a) Shot gather #1 from 21YY. (b) Removal of high-frequency random noise and coherent linear noise. (c) Application of a frequency-offset coherent noise filter and tau-p linear noise attenuation for surface wave removal. (d) Result after applying pre-stack time migration.**

86

87

88

89

90

91

**References**

93   Hildebrand, S. T.: Two representations of the fan filter, Geophysics, 47, 957–959, https://doi.org/10.1190/1.1441363, 1982.

94   SLB: Omega Geophysical Processing System – Anomalous Amplitude Noise Attenuation (ANOMALOUS_AMP_ATTEN)
95   Module Documentation, Version 27.14, SLB manual, Houston, TX, 2025a.

96   SLB: Omega Geophysical Processing System – Curvelet Transform (CURVELET_TRANSFORM) Module Documentation,
97   Version 22.1, SLB manual, Houston, TX, 2025b.

98   SLB: Omega Geophysical Processing System – Surface-Consistent Deconvolution Analysis (SC_DCN_SPCTRL_ANL)
99   Module Documentation, Version 13.18, SLB manual, Houston, TX, 2025c.

100  SLB: Omega Geophysical Processing System – F-X Coherent Noise Suppression (FXCNS) Module Documentation, Version
101  3.16, SLB manual, Houston, TX, 2025d.

102  Yilmaz, Ö.: Seismic data analysis: Processing, Inversion, and Interpretation of Seismic Data, Appendix B.8 (Surface-
103  Consistent Deconvolution), Society of Exploration Geophysicists, 262–266 pp., 2001.

---

## Author Response (AR3)

**Editor**

L224: "approximately 2.01 m to 5.27 m." Quoting these values to two decimal places doesn't sound very approximate! Might it be better to report them as "approximately 2.0 m to 5.3 m"?

➜ (Line 224) We have revised "approximately 2.01 m to 5.27 m" to "approximately 2.0 m to 5.3 m".

Figure 6: label "Discontinous" should be "Discontinuous"

➜ (Figure 6 and 9) We have revised "Discontinous" to "Discontinuous".
➜ (Figure 8) The title for the legend has been added to address the previous omission.

L372-373, L377: is precision of two decimal places really justified for these errors? Would integers be more defensible?

➜ In accordance with the editor's suggestion, values have been rounded to the nearest integer. Given the importance of data reliability, the confidence interval has been extended to 99%, and the updated results are provided in Supplementary information S3.
➜ (Supplementary information S3)

S3. Seismic data acquisition

1. IPR data uncertainty in glacier thickness estimates

The vertical uncertainty of the GNSS data was estimated to be 0.98 m. At the 99% confidence level, the uncertainties in surface and bed elevation measurements were calculated as ±6.9 m and ±32.7 m, respectively (Figure S5). When incorporating the GNSS vertical uncertainty, the total uncertainties in surface and bed elevations increased slightly to ±7.0 m and ±32.7 m, respectively. Consequently, the overall uncertainty of the IPR-derived ice thickness was estimated to be ±33.4 m.

$$U_{IPR} = \sqrt{(\pm 7.0 \text{ m})^2 + (\pm 32.7 \text{ m})^2} = \pm 33.4 \text{ m}. \hspace{2cm} \text{S3.1}$$

[Figure]

**Figure S1. Measurement uncertainty results of surface and ice bottom elevations from IPR data.**

2. Seismic data uncertainty in glacier thickness estimates

The uncertainty associated with seismic picking arises from the presence of noise and is quantified as picking uncertainty ($U_{pick}$), which can be estimated using the following equation (Abakunov et al., 2020):

$$\sigma_{pick} = 2\sqrt{\frac{T^2}{4\pi^2}\frac{1}{SNR}}, \qquad\qquad \text{S3.2}$$

where $T$ is the period of the central frequency and SNR is the signal-to-noise ratio. Based on this formulation, the SNRs of the first arrival and the ice–water interface signals were determined to be 88.8 dB and 10 dB, yielding vertical uncertainties of 0.2 m and 2.1 m, respectively.

$$U_{pick} = 2.58\,\sigma_{pick} = \pm5.4 \text{ m.} \qquad\qquad \text{S3.3}$$

The total picking uncertainty at the 99% confidence level is 5.4 m. Under glacial temperature conditions of –2 ± 2 °C, where the average P-wave velocity in ice is $3800 \pm 5$ m s⁻¹ (Kohnen, 1974), the uncertainty associated with the variability in seismic velocity is estimated to be ±5.3 m. Assuming a firn layer thickness of 100 m, the combined measurement uncertainty of the seismic results at the 99% confidence level is calculated to be 7.6 m.

$$U_{Seis} = \sqrt{(\pm5.4 \text{ m})^2 + (\pm5.3 \text{ m})^2} = \pm7.6 \text{ m.} \qquad\qquad \text{S3.4}$$

**Referee #2**

One comment that I think may be worth addressing is the consistent appearance of the seismically derived bed profile beneath the radar derived depth. Is this simple an effect of dominant wavelength? Are the boundaries two slightly different physical reflectors? Something to think about, and maybe add a sentence about.

➜ As noted by the reviewer, the ice thickness estimated from radar data is consistently greater than that from seismic data across most of the survey area. This discrepancy may result from an overestimation of the radar wave velocity (0.169 m/ns) used in the ice. Although Ju et al. (2025) adopted this representative value from the literature (Reynolds, 2011), it may differ from the actual radar wave speed in the study area. Additionally, the uncertainty in ice bottom picking from the radar data is relatively large (±32.7 m), and this must be considered when comparing the two datasets. Nevertheless, the consistency between the two datasets remains within the bounds of uncertainty, supporting their mutual reliability. This consistency further validates the integrated application of both methods for characterizing subglacial lake environments.
(Lines 380-385) We have added the sentence "The ice thickness derived from radar data is generally greater than that obtained from seismic data across most areas. This discrepancy may be attributed to an overestimation of the radar velocity. Ju et al. (2025) adopted a commonly used literature-based radar velocity of 0.169 m/ns, which may differ from the actual radar velocity in the study area. Additionally, the uncertainty in measuring the ice bottom in the radar data is ±32.7 m, and this must be considered when comparing the two datasets. Despite these factors, the two datasets show a high level of consistency within the uncertainty bounds. This consistency supports the mutual reliability of both methods and validates their integrated application for subglacial lake characterization.".

L82: qualitatively? aren't matching absolute amplitudes

➜ In land-based seismic data, the amplitude of each shot gather can vary due to several factors. When integrating multiple shot gathers during data processing, amplitude normalization is essential. As shown in the workflow (Fig. 5), anomalous amplitude attenuation was applied to suppress outlier signals by interpolating between adjacent traces.

L112: how do you know it's cyclic as opposed to simply a single drainage/refilling event?

➜ We thank the reviewer for identifying this potential oversight. Given the limited duration of monitoring, there is currently insufficient evidence to confirm the periodicity of the observed surface elevation changes. Accordingly, the sentence has been revised as follows:
(Line 112) We have revised the sentence "These patterns of elevation change strongly suggest that SLD2 is an active subglacial lake, **with cyclic** drainage and refilling likely contributing to the presence of subglacial sediments (Siegfried et al., 2023)." to
"These patterns of elevation change strongly suggest that SLD2 is an active subglacial lake, **and that such** drainage and refilling are likely contributing to the presence of subglacial sediments (Siegfried et al., 2023)."

L169: refracted vs pseudoacoustic? difference in terminology ?

➜ To avoid potential misunderstanding, we have clarified that the term "refracted wave" refers to the apparent velocity derived from the first arrivals in the raw shot gather data. The revised sentence emphasizes that both the direct and refracted wave velocities were empirically measured from the observed travel-time curves.
(Lines 168-171) We have revised the sentence "In these shot gathers, the velocity of the direct wave is estimated to be approximately 1800 m/s, and the refracted wave velocity in firn-ice transition is

approximately 3800 m/s" to

"In these shot gathers, both the direct wave and the refracted wave velocities were derived from first-arrival travel-time analysis. The direct wave velocity was estimated to be approximately 1800 m s$^{-1}$, while the higher-velocity arrival—interpreted as a refracted wave traveling through the firn–ice transition zone—exhibited an apparent velocity of approximately 3800 m s$^{-1}$".

L223: would these be due to crevasses or topography variation? simply filtering artifacts? what is the reason for spurious arrivals?

➔ The seismic velocities cited in this sentence correspond to the minimum and maximum P-wave velocities of water, firn, and ice. The sentence has been revised to eliminate any potential ambiguity. (Line 225) Assuming seismic wave velocities between 1396 m s$^{-1}$ **(water)** and 3800 m s$^{-1}$ **(ice)**, the corresponding vertical resolutions, which are calculated using the quarter-wavelength criterion, range from approximately 2.0 m to 5.3 m.

Figure 6: why is the (③) reflection in 21Y so much more geometrically complex than the glacier-lake interface

➔ The geometric complexity of reflection (③) in line 21Y, as compared to the glacier–lake interface (①), is attributed to the combined influence of multiple signal interferences. Specifically, the discontinuous character of reflection (①) and the presence of a ghost reflection (②) create overlapping waveforms that interfere with the interpretation of reflection (③). This interference results in apparent geometric irregularities that may not reflect actual subsurface features. Consequently, it remains unclear whether reflection (③) corresponds to a true subglacial interface—such as a sedimentary boundary—or whether it is an artifact of waveform interference. To reduce this interpretational uncertainty, we are currently preparing to apply deghosting techniques in future data processing. This additional step is expected to enhance the clarity of complex reflection patterns, particularly in regions such as reflection (③), where distinguishing structural boundaries is challenging. The refined results will be presented in subsequent studies.